# Costless metabolic secretions as drivers of interspecies interactions in microbial ecosystems

Alan R. Pacheco [1], Mauricio Moel[2] & Daniel Segrè [1,2,3,4]

Metabolic exchange mediates interactions among microbes, helping explain diversity in microbial communities. As these interactions often involve a fitness cost, it is unclear how stable cooperation can emerge. Here we use genome-scale metabolic models to investigate whether the release of "costless" metabolites (i.e. those that cause no fitness cost to the producer), can be a prominent driver of intermicrobial interactions. By performing over 2 million pairwise growth simulations of 24 species in a combinatorial assortment of environments, we identify a large space of metabolites that can be secreted without cost, thus generating ample cross-feeding opportunities. In addition to providing an atlas of putative interactions, we show that anoxic conditions can promote mutualisms by providing more opportunities for exchange of costless metabolites, resulting in an overrepresentation of stable ecological network motifs. These results may help identify interaction patterns in natural communities and inform the design of synthetic microbial consortia.

[1] Graduate Program in Bioinformatics and Biological Design Center, Boston University, Boston, MA 02215, USA. [2] Department of Biology, Boston University, Boston, MA 02215, USA. [3] Department of Biomedical Engineering, Boston University, Boston, MA 02215, USA. [4] Department of Physics, Boston University, Boston, MA 02215, USA. Correspondence and requests for materials should be addressed to D.S. (email: dsegre@bu.edu)

The astonishing number of microbial species observed in nature[1–3] seems to contradict classical ecological theory, which predicts far less biodiversity in many nutrient-poor environments[4,5]. A variety of explanations have been proposed as possible solutions to this discrepancy[6–10], including metabolic cross-feeding[11–13]. This phenomenon, in which one species produces metabolites that are then consumed by another, has been shown to enhance the capacity of microbes to survive in resource-poor environments[14–16]. However, it is not clear how these cooperative phenotypes emerge, as they often involve the exchange of metabolites that are costly for the producer. This apparent altruism introduces the potential for the rise of cheating organisms that do not contribute common goods but still benefit metabolically from others, challenging community stability[17]. Previous studies have addressed this dilemma in different ways[18–21], though some modes of exchange are not associated with a drop in fitness[22]. Given the evolutionary dilemmas associated with costly cooperation, we ask here whether the exchange of metabolic secretions that do not impose an effective fitness cost can in principle help account for the degrees of biodiversity observed in nature.

It is known that microbes often secrete waste products (e.g., *Escherichia coli* secreting acetate under limited oxygen) that can support other species[15]. This phenomenon may allow community benefits to emerge as a product of otherwise selfish acts by individual organisms. It is not obvious, however, whether such behavior extends beyond a few fermentation byproducts or how widely it varies across microbial species and environmental contexts[23]. Moreover, it is not clear whether these byproducts have the potential to enable or enhance growth of other species in complex communities.

Here, we use computational metabolic modeling to explore the environmental modifications brought about by "costless" metabolite secretion, as well as the interspecies interactions enabled by this type of exchange. For our purposes, we define a metabolite as costless if the predicted growth rate of an organism secreting that metabolite is not less than its growth rate when the metabolite is not secreted. As we will illustrate in detail, the costless nature of a given metabolic secretion is strongly dependent on environmental conditions and may hinge on the techniques used to generate predictions. Our computational framework is based on flux balance analysis (FBA)[24], which we use to predict the growth phenotypes and beneficial interactions mediated by costless metabolites for 24 microbial species under a large set of carbon source combinations. Through this method, we obtain a global view of cross-feeding opportunities that can mediate the emergence of beneficial interactions and the maintenance of biodiversity in natural communities.

Previous computational models of microbial community metabolism have yielded useful predictions on mutually beneficial and competitive behaviors[14,16,25–29]. In this study, we evaluate the impact of costless metabolic secretions by applying FBA to a large space of microbial species and environmental conditions. Moreover, we quantify the degree to which environmental variables, such as oxygen and carbon source, contribute to costless metabolic secretions, interspecies interactions, and community stability. Though the present work focuses on secretions and interactions predicted computationally, we restricted our analysis to microbes associated with high-quality and experimentally verified in silico models, which have allowed us to make predictions consistent with previously established empirical knowledge. However, the current analysis should be viewed as an exploration of a large space of stoichiometrically possible costless interactions (inscrutable to such an extent at the experimental level), whose global patterns can motivate and inform future experimental and theoretical endeavors.

## Results

**Metabolite secretion cost depends on environmental context.** Understanding whether the secretion of a metabolite by an organism is associated with a decrease in fitness (interpreted here as growth rate) is difficult to assess experimentally, but can be readily calculated using genome-scale models of metabolism (see Methods). For example, one can impose the secretion of a given compound at a given rate $v_s$, and then ask whether this constraint is expected to cause the organism's growth rate ($v_{g,s}$) to be less than its growth rate without this constraint ($v_{g,0}$). A small set of simulations of this kind for a single organism (Supplementary Fig. 1) exemplifies the spectrum of possible outcomes: depending on the carbon sources provided, different metabolites can be produced either (i) at the expense of growth capacity, (ii) with no apparent effect, or (iii) even to its benefit. For cases (ii) and (iii), since fitness is not reduced by the metabolite secretion, $v_{g,s} \geq v_{g,0}$. The existence of solutions that satisfy this equation (i.e., flux states that have higher growth in presence of a metabolic secretion) forms the basis of our subsequent calculations. The above equation can thus be viewed as the defining characteristic of a costless metabolic secretion.

We note that while FBA provides mechanistic insight into the tradeoff between metabolic reaction costs and benefits, current predictions (including those on the costless nature of a metabolic secretion) may depend on factors not captured by our method, including temperature[30], signaling and gene regulation[31–35], pH changes[36], and explicit pathway-dependent cost of enzyme production[37]. Our definition of "costless" may therefore be interpreted as a heuristic that captures expected spontaneous metabolite secretions, in contrast to secretion that would be associated with a growth or fitness reduction.

**Costless secretions promote environmental enrichment.** Having illustrated in an individual case how metabolite secretion costs can strongly depend on carbon sources, we sought to map the prevalence of costless secretions across a broad set of organisms and environments. As an initial core analysis, we carried out a total of 1,051,596 unique simulations, each with two organisms from a set of 14 genome-scale models of facultative anaerobes and two carbon sources from a set of 108 compounds (Fig. 1, Supplementary Data 1, Supplementary Table 1). We chose facultative anaerobes in order to enable a direct comparison of secretion profiles in oxic vs. anoxic conditions. Each simulation was conducted as an iterative process that emulates a coculture experiment, uniquely defined by the organisms involved, the carbon sources provided, and the availability of oxygen. At each iteration, we used FBA to determine the ability of each organism to grow on the provided medium, in addition to the set of metabolites predicted to be spontaneously (i.e., costlessly) secreted by each microbe. We incorporated several measures into these simulations to minimize false-positive reporting of secreted metabolites (see Methods).

If at the first iteration ($c = 1$, Fig. 1) at least one organism was able to grow on the carbon sources provided, all newly secreted costless metabolites were added to the medium for the next iteration. This process was repeated until no new metabolites were produced (defining a final iteration $c = c_s$). Upon running such simulations for all combinations of species and environments, we obtained distributions for the value of $c_s$ (Fig. 2a). A majority of cases reached a steady state after only one iteration, possibly due to organisms secreting multiple byproducts that only contributed weakly to subsequent secretions.

In aggregate, our simulations showed a rightward shift in the diversity of metabolites secreted under anoxic conditions when compared to the number secreted when oxygen was available

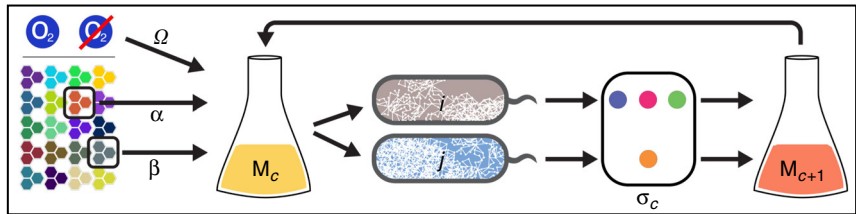

**Fig. 1** Simplified schematic illustrating the in silico experiments. A growth medium ($M_c$) containing two carbon sources ($\alpha$, $\beta$) with or without oxygen ($\Omega$) is provided to genome-scale metabolic models of two microbial organisms ($i$, $j$). If at least one organism grows, any costlessly secreted metabolites ($\sigma_c$) are added to the medium, which is fed back to the organisms. This process is repeated for a series of iterations $c$, and terminates at iteration $c_s$, defined as the last iteration in which any new metabolites were secreted into the medium

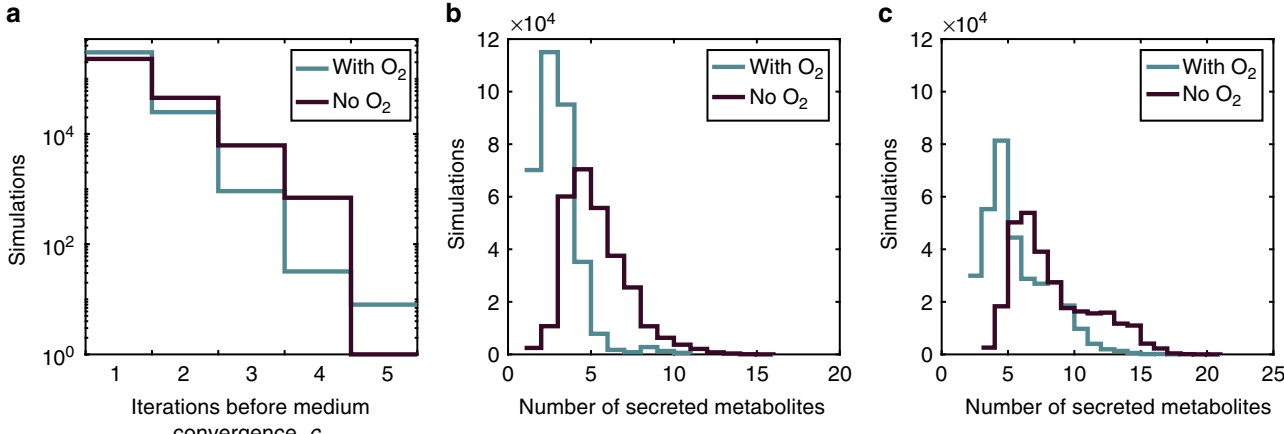

**Fig. 2** Analysis of costlessly secreted metabolites in pairwise simulations. Only simulations that led to growth of at least one organism are shown. **a** Distribution of number of expansions until final medium expansion iteration. Ninety-two percent of simulations reached a steady medium composition after only one iteration with oxygen, compared to 82% of simulations without oxygen. **b** Distribution of the number of metabolites secreted into the medium by one or both organisms in a pair after one iteration of FBA ($c = 1$). These distributions were unimodal for both conditions, centered between two and three metabolites with oxygen and around five metabolites without oxygen. After this first iteration, the maximum number of secreted metabolites was 11 with oxygen and 16 without oxygen. In the anoxic simulations, the central carbon metabolites most commonly secreted after the first iteration were fermentation byproducts such as acetate, formate, succinate, and ethanol. These metabolites were secreted in 87.5%, 74.5%, 25.7%, and 20.2% of growth-yielding simulations respectively. With oxygen, the most commonly secreted central carbon metabolites after the first iteration were formate and acetate, secreted in 46.8% and 18.3% of growth-yielding simulations, respectively. **c** Distribution of the number of metabolites secreted by one or both organisms after the last iteration of FBA ($c = c_s$). The last iteration is defined as the iteration in which no additional metabolites were released into the medium. The total number of secreted metabolites followed similar distributions with a maximum at 18 and 21 metabolites for oxic and anoxic conditions, respectively. Despite the large variability in number of expansions and number of secreted metabolites, we observe a poor correlation between these distributions, indicating that a simulation resulting in a high number of expansions does not necessarily result in a high number of metabolites being secreted (Supplementary Fig. 3)

(Fig. 2b), as well as a shift in the quantity of metabolites secreted between the first and last iteration of each simulation (Fig. 2c). This latter effect reflects organisms taking up metabolites secreted by themselves or their partner, and secreting different metabolites as a response. Based on these results, we hypothesized that oxygen availability would be among the best indicators of the metabolites secreted in a simulation. To quantify this effect, we applied a machine learning approach to a modified simulation set consisting of all 14 organisms individually feeding on a single carbon source (see Methods). Using this method, we found that sets of secreted metabolites could be used to yield varying degrees of information on simulation starting conditions. Specifically, oxygen availability, species identity, and carbon source type could be predicted with cross-validation accuracies of 93.4%, 58.0%, and 85.3%, respectively. Notably, organism identity appeared to be not strongly associated with specific costless secretions compared to carbon source and oxygen. This may be due to the fact that, while an organism may have a pathway to secrete a particular byproduct, utilization of that pathway would be strongly contingent on the presence of the

necessary substrates. The observed associations of secretions with the carbon source mirrored previous experimental observations, which identified carbon sources as the main drivers of community composition through metabolic cross-feeding[13]. While the specific concentration of environmental substrates could in principle affect predicted secretions, we found through dynamic FBA[38,39] (dFBA) simulations that substrate concentration had a negligible effect on the identity of secreted metabolites (see Methods and Supplementary Data 2). Nonetheless, one should consider the possibility that this result may be due to limitations of constraint-based modeling, which may be overcome in future studies.

**Useful costlessly secreted byproducts are abundant**. Our analysis revealed that most organisms secreted a broad distribution of metabolically useful compounds without cost in a variety of environmental conditions (Fig. 3, Supplementary Fig. 4a). Though inorganic compounds such as water and carbon dioxide were, as expected, the most commonly secreted molecules across

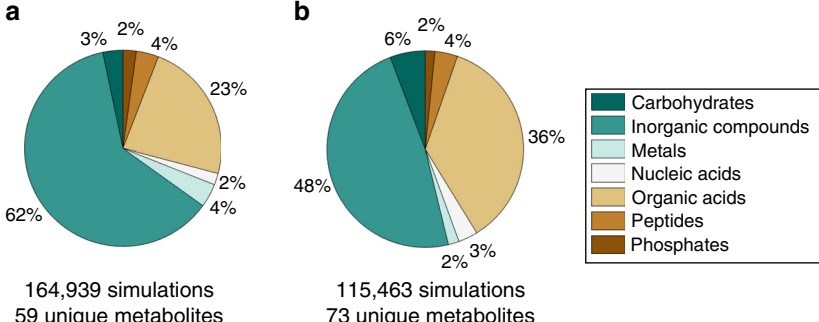

**Fig. 3** Categorization of metabolites secreted costlessly in all simulations. **a**, **b** Categorization for all simulations with oxygen (**a**) and without oxygen (**b**). Though inorganic waste products (e.g., water, $CO_2$) make up the majority of unique metabolites secreted with and without oxygen, release of potentially valuable metabolites such as organic acids, carbohydrates, and peptides is observed in a major subset of simulations. In anoxic simulations in which at least one organism fully reduced nitrate into nitrogen gas, we observed a modest reduction in the number of fermentation byproducts secreted ($2.81 \pm 1.11$ metabolites for non-nitrate respirers vs $2.38 \pm 0.54$ metabolites for nitrate respirers)

all simulations, nitrogen-containing compounds such as nitrite, ammonium, and urea were secreted in 73.5% of the analyzed cases, suggesting maintenance of an appropriate carbon-to-nitrogen ratio in the cell. We note specifically that nitrite is secreted in fewer than 100 simulations with oxygen, but its secretion is prevalent in anoxic simulations—a phenomenon previously observed in anaerobic enteric bacteria[40]. Moreover, ~10% of anoxic simulations resulted in at least one organism fully reducing nitrate into nitrogen gas, suggesting that anaerobic respiration was a preferred strategy in some environments. Organic acids made up the second most abundant category of costless secretions, followed by nucleotides, peptides, and carbohydrates. Altogether, this space of secreted metabolites points to a large variety of molecules that can be freely produced in resource-poor environments.

Despite our careful design of the simulation process, it remains difficult to quantify the degree to which these secretions will be observed experimentally. For this reason, we have relied exclusively on genome-scale metabolic models that have undergone experimental validation under conditions that in many cases mirror those that we have simulated, in addition to imposing our own set of constraints (see Methods). Moreover, though empirical testing of every simulation we performed is inaccessible, we note that experimental data from previously published work supports key portions of our predictions (Supplementary Table 4). In an additional effort to ensure the accuracy of the set of secreted metabolites, we also carried out all simulations using alternative objective functions. In particular, though optimization of growth reflects the possibility of organisms "selfishly" growing as rapidly as possible and "unintentionally" secreting useful metabolites, alternative objective functions may best capture metabolic regimes relevant across different conditions. We therefore compared metabolite secretion profiles inferred by maximizing growth to those obtained through minimization of biomass production, as well as maximization and minimization of ATP production. All objectives gave rise to secretion profiles highly similar to each other, with an increase of only 0.18% of all predicted metabolic secretions in the growth maximization condition relative to the others (see details and results in Supplementary Fig. 5, and Supplementary Table 5).

Given the abundance of secretions from different organisms, we asked whether specific metabolite secretions were highly correlated. We thus used a Spearman correlation analysis to identify secretion patterns that appeared with high frequency (Supplementary Fig. 6). In the presence of oxygen, we observed a strong co-occurrence of glycerol, lactate, succinate, malate, and acetate, which may reflect the high frequency of secretion of these

carbon-containing compounds. We also observed positive, but weaker correlations between these metabolites and other central carbon compounds such as fumarate, citrate, and 2-oxoglutarate. Our analysis also pointed to the simultaneous release of multiple nitrogen-containing compounds, chiefly urea, ammonium, and nitrate. Without oxygen, we observed stronger correlations between secretion of nitrogen-containing compounds and fermentation byproducts. Amino acids also co-occurred with high frequency without oxygen in patterns consistent with examples of previously studied exometabolomic profiles, including those showing co-secretion of central carbon intermediates in *E. coli* and of amino acids in yeast[41,42], as well as time-dependent patterns of metabolites released simultaneously in soil communities[43]. These co-secretion profiles suggest that environments modified by metabolic activities of existing organisms may be simultaneously enriched by specific combinations of molecules.

Having mapped the space of metabolites secreted at no fitness cost to the producer, we sought to understand which metabolites could be subsequently taken up by other organisms. We found that the organic metabolites most commonly exchanged across species were central carbon intermediates, secreted mainly in anoxic conditions (Supplementary Fig. 4b). These secretion patterns mirrored those of anoxic gut bacteria, which divide the task of digesting complex polysaccharides by exchanging intermediate organic acids[11,44]. Importantly, we observed that amino acids, secreted chiefly by *Saccharomyces cerevisiae*, but also in a substantial number of simulations by *Salmonella enterica*, *Klebsiella pneumoniae*, and *E. coli*, were among the most frequently exchanged costless metabolites. This phenomenon has been previously documented in relation to overflow metabolism in *S. cerevisiae*[45] and *E. coli*[46,47], as well as in yeast–bacteria symbioses[48,49]. This high prevalence of exchange underscores the metabolic utility of these secreted byproducts.

**Costless metabolite exchange enhances growth capabilities.** We next assessed how often the exchange of costlessly produced molecules could directly enable growth of other organisms that would otherwise not grow on the initial environmental nutrients. Before taking into account the costless secretions, 18.2% and 11.9% of simulations predicted growth of both organisms with and without oxygen, respectively (Fig. 4a). After the organism pairs were allowed to exchange costlessly secreted metabolites, our algorithm predicted a substantial increase in growth-supporting environments (72.7% with oxygen and 82.5% without oxygen relative to minimal medium), suggesting that exchange of costlessly secreted metabolites can enable growth of additional organisms in resource-poor environments.

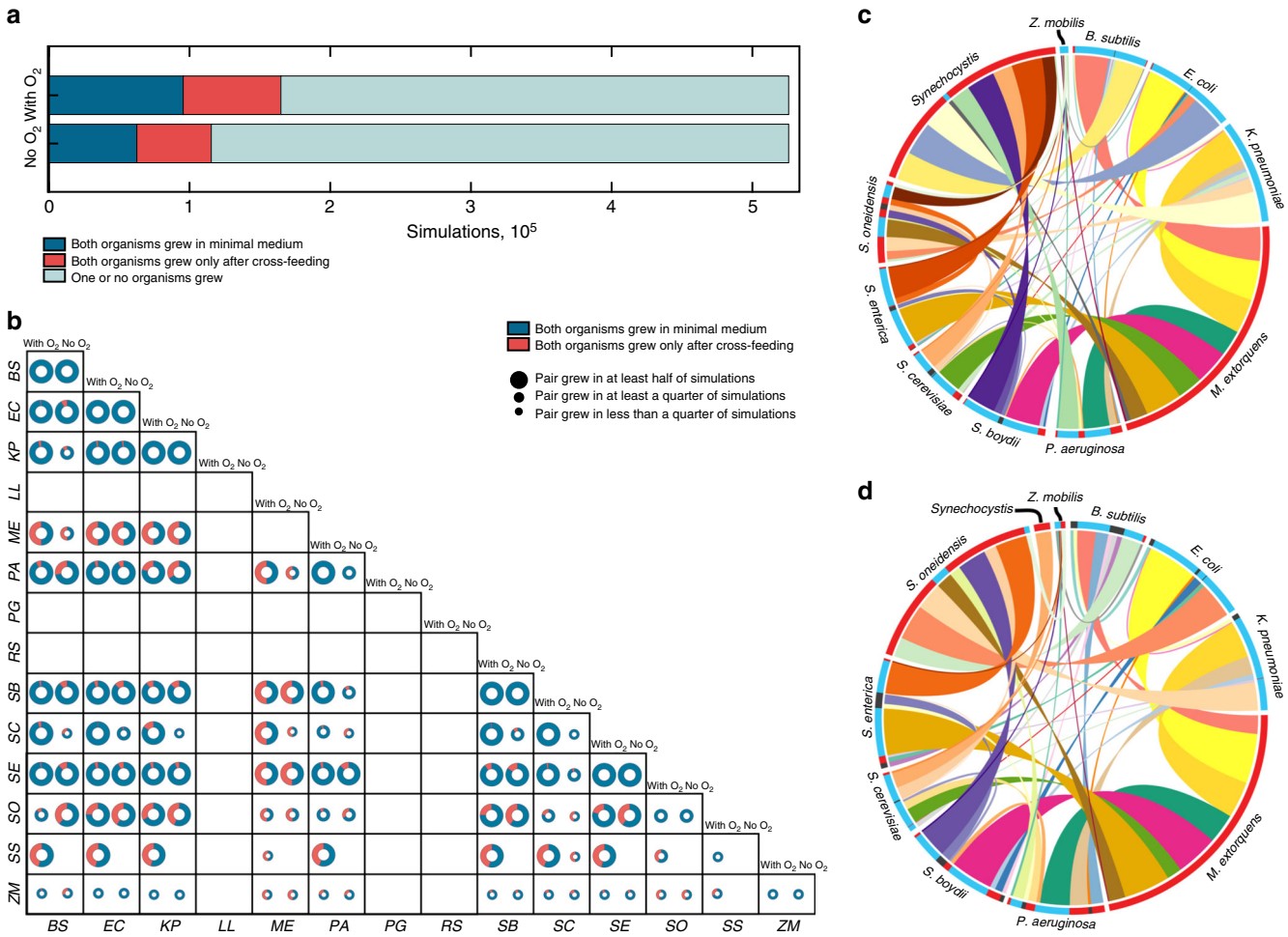

**Fig. 4** Growth outcomes of pairwise cross-feeding simulations. **a** Growth outcomes of all in silico experiments with and without oxygen, grouped by pairwise growth phenotype. **b** Organism-specific growth outcomes. Size of circles represent the relative number of environments in which an organism was able to grow out of 5774 in silico experiments with each partner. **c, d** Frequency of obligate pairwise growth by species in single carbon source simulations for oxic ($N = 69{,}420$, **c**) and anoxic ($N = 52{,}897$, **d**) conditions. Each color ribbon is unique to an individual species pair. Width of ribbons is proportional to the number of experiments in which obligate syntrophy was predicted for each species pair. Radial axis colors represent directionality of exchange: blue: organism provided essential metabolites to partner organism in over 75% of simulations; red: organism received essential metabolites in over 75% of simulations; gray: both organisms gave and received essential nutrients in most simulations. Most pairings of organisms were imbalanced, with one organism more frequently providing essential nutrients to another. For example, with oxygen, *Synechocystis* relied on metabolites from nine different organisms across the vast majority of simulations in which it grew with a partner. As all organisms were grown heterotrophically, carbon dioxide and ammonium were the main byproducts that enabled growth of *Synechocystis* in these simulations. BS: *B. subtilis*, EC: *E. coli*, KP: *K. pneumoniae*, LL: *L. lactis*, ME: *M. extorquens*, PA: *P. aeruginosa*, PG: *P. gingivalis*, RS: *R. sphaeroides*, SB: *S. boydii*, SC: *S. cerevisiae*, SE: *S. enterica*, SO: *S. oneidensis*, SS: *Synechocystis*, ZM: *Z. mobilis*

In addition to a global increase in growth capabilities due to costless metabolite secretion, we observed species-specific growth patterns that varied widely across our dataset (Fig. 4b). *Lactococcus lactis* and *Porphyromonas gingivalis*, for example, are host-associated microbes that are auxotrophic for a wide range of metabolites and that often depend on metabolic products from the host or other commensal microbes[50,51]. In our study, these organisms failed to grow in all simulations even after costless metabolites were made available by a partner. This failure to sustain growth of highly dependent organisms suggests that there is an upper limit to the degree to which costless metabolite production can enable species growth, especially in the minimal environments that were tested. Nonetheless, most of the metabolites that these organisms require to grow were producible separately by multiple species, suggesting a possible important role of multi-partner cross-feeding interactions in complex communities. Aside from these extreme cases, our analysis shed

light on the performance of generalist organisms, such as *E. coli*, *K. pneumoniae*, *S. cerevisiae*, and *S. enterica*. These organisms grew in at least half of all tested environmental conditions, in contrast with organisms such as *Methylobacterium extorquens* or *Zymomonas mobilis*, which exhibited much more limited pairwise growth capabilities. The growth patterns of these latter organisms suggest a greater dependence on the metabolic byproducts of their partners, particularly in anoxic conditions.

As our study relied on a limited set of curated metabolic models, we wondered how sensitive these results were to the organisms being assessed. In order to explore possible bias, we conducted additional simulations in which we binned organisms by environmental habitat. These simulations were separated into three sets: the first with 13 aquatic microbes grown aerobically, the second with 12 soil microbes grown aerobically, and the third with 12 human gut-associated microbes grown anaerobically. These simulations employed additional genome-scale models

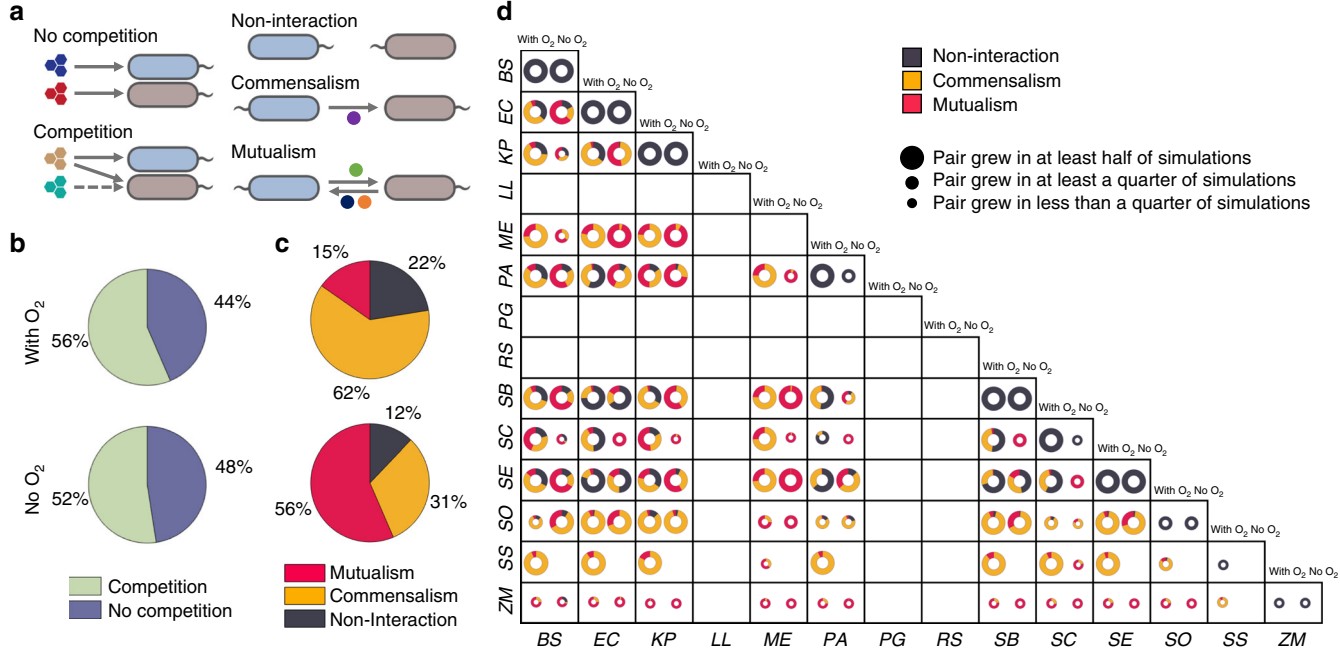

**Fig. 5** Distribution of metabolic interaction types. **a** Schematic representation of interaction types arising from costlessly secreted metabolites. Competition is defined as both organisms consuming the same carbon source. Commensalism is defined as a unidirectional exchange of one or more costlessly secreted metabolites, and mutualism is defined as a bidirectional exchange of one or more costlessly secreted metabolites. **b** Overall distributions of competitive and non-competitive interactions for oxic (out of 164,939 simulations that yielded pairwise growth) and anoxic conditions (out of 115,463 simulations that yielded pairwise growth). **c** Overall distributions of general interactions mediated by costless metabolites for oxic and anoxic conditions. These interactions at the level of secreted metabolites exist simultaneously with competition or no competition for a primary carbon source. **d** Organism-specific growth outcomes and interaction type distributions. Size of circles represent the relative number of environments in which an organism was able to grow out of 5774 in silico experiments with each partner. BS: *B. subtilis*, EC: *E. coli*, KP: *K. pneumoniae*, LL: *L. lactis*, ME: *M. extorquens*, PA: *P. aeruginosa*, PG: *P. gingivalis*, RS: *R. sphaeroides*, SB: *S. boydii*, SC: *S. cerevisiae*, SE: *S. enterica*, SO: *S. oneidensis*, SS: *Synechocystis*, ZM: *Z. mobilis*

(including obligate aerobes or obligate anaerobes, see Supplementary Data 1) that were not used in our core analysis of 14 facultative anaerobes. By analyzing the expanded set of organisms in a habitat-specific manner, we found that exchange of costless metabolites substantially improved the ability of minimal environments to support pairwise growth in all three habitats (Supplementary Fig. 7a). Notably, metabolite secretion and exchange for aquatic and soil microbes resembled the profiles found for the core organisms grown with oxygen (Supplementary Fig. 7b, c, Supplementary Table 6). Conversely, the distribution of secreted metabolites for gut-associated microbes featured widespread secretion and exchange of organic acids that were similar to those found across core organisms grown anoxically (Supplementary Fig. 7d, Supplementary Table 6).

**Costless metabolic exchange yields specific partnerships.** After analyzing general growth outcomes across our entire simulation set, we sought to determine which specific organisms could not grow without the costless secretions of a partner. Our simulations identified a diverse space of such organisms, with most species exhibiting at least one case of obligate syntrophy with all others (Fig. 4c, d). Though many organisms had balanced distributions of dependence (i.e., organism *i* enabled the growth of organism *j* in some cases, and organism *j* enabled the growth of *i* in others), the majority of such relationships were unidirectional. One striking example of this phenomenon is that of cyanobacteria and heterotrophic organisms, with *Synechocystis* depending frequently on other organisms. We also observed that *E. coli*, *Bacillus subtilis*, and *S. cerevisiae*, three species commonly used as model microbial organisms, were more frequently the giving organisms in cases of obligate syntrophy. These pairings not only shed light

on the mechanisms behind interspecies codependencies, but may also serve as a map for assembling co-dependent synthetic communities stabilized by costless metabolic exchange.

**Carbon sources exhibit cooperativity in promoting growth.** In addition to characterizing the global space of in silico growth phenotypes, we examined how cooperativity of carbon sources could enhance growth capabilities in organism pairs. Drawing from techniques used to quantify epistatic interactions[52], we defined the cooperativity index *C* of two carbon sources α and β as the difference between the number of simulations that resulted in growth from both carbon sources ($g_{\alpha,\beta}$) and the product of the number of simulations that resulted from single carbon sources ($g_\alpha$, $g_\beta$). These counts were normalized by the total number of simulations involving the specific pairing of carbon sources being analyzed (represented here by the combinatorial formula $\binom{N}{2}$), as follows:

$$C^{\alpha,\beta} = \frac{g_{\alpha,\beta}}{\binom{N_{\alpha,\beta}}{2}} - \left( \frac{g_\alpha}{\binom{N_\alpha}{2}} \times \frac{g_\beta}{\binom{N_\beta}{2}} \right). \qquad (1)$$

This metric reflects the cooperative potential of each carbon source pair relative to that of each carbon source in isolation. Upon averaging a single carbon source over its cooperativity index, we obtain a relative degree to which a carbon source "depends" on another to sustain growth. By framing cooperativity in this context, we observed that simple sugars such as glucose

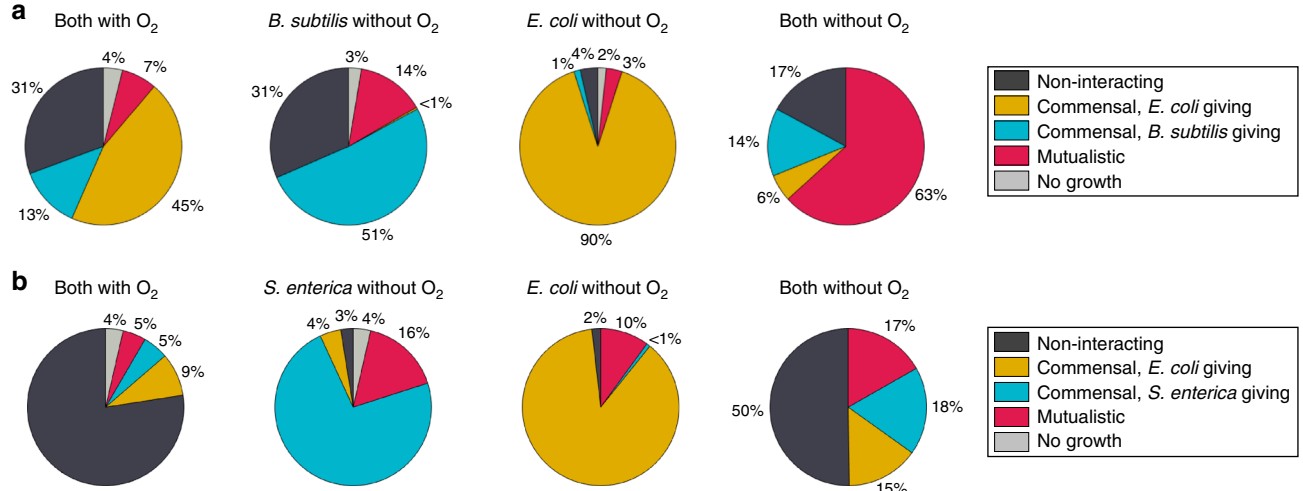

**Fig. 6** Interaction type distributions from hybrid oxic–anoxic simulations for two organism pairs. **a** *E. coli* with *B. subtilis*. **b** *E. coli* with *S. enterica*. Both hybrid simulations demonstrate that regardless of organism, availability of oxygen is a strong determiner of the potential for bidirectional exchange. When oxygen is not provided to an organism in these simulations, it tends to provide metabolites to its partner, resulting in an abundance of commensal interactions. These scenarios may act as a "stepping stone" toward fully anoxic environments, in which mutualistic interactions become more prevalent

and sucrose had relatively low cooperativity indices, that is, they were able to sustain growth efficiently on their own. In contrast, more complex molecules and dipeptides had higher average cooperativity indices, indicating they performed better in the presence of another carbon source. We grouped these average cooperativity indices through hierarchical clustering (Supplementary Fig. 8, Supplementary Table 7) and observed general clustering by carbon source type—especially with sugars and amino acids appearing in distinct groups. This analysis illustrates the nonlinear effects of adding additional nutrients to a minimal medium, underscoring the observed complex metabolite usage patterns in organism pairs.

**Costless cross-feeding can offset competition for nutrients.** Our analysis so far has examined the contexts in which a metabolite can be secreted costlessly, as well as the potential for these metabolites to promote growth. Additional insight about the relevance of these interactions can be obtained by comparing them to ecological expectations of cooperation and competition. Towards this goal, we defined six types of possible interactions: non-interaction, commensalism (unidirectional exchange), and mutualism (bidirectional exchange), each with or without competition for a primary carbon source (Fig. 5a). We chose to decouple competition for nutrients from exchange of secreted metabolites in order to more fully understand the degree to which the latter can promote organism coexistence despite resource scarcity. When analyzing our dataset under this framework, we found that competition for one or both carbon sources constituted the majority of all interactions (Fig. 5b), as previously observed experimentally[53]. However, these predicted competitive phenotypes were observed to frequently occur simultaneously with potentially beneficial interactions mediated by metabolic byproducts.

Our modeling predicted bidirectional interactions to be far more common without oxygen than with oxygen (Fig. 5c). We obtained a more fine-grained perspective on costless metabolic interactions by considering the distributions of interaction types by species pairs (Fig. 5d). For example, the majority of pairings of *M. extorquens* with *B. subtilis*, *E. coli*, and *K. pneumoniae* exhibited commensal interactions (chiefly with *M. extorquens* receiving). In contrast, the distribution of interactions shifted

toward mutualism when oxygen was made unavailable. These patterns were also mirrored in a majority of individual species pairings. As with the positive shift observed in the distributions of secreted metabolites (Fig. 2b, c), we attributed the increased prevalence of mutualistic interactions without oxygen to a greater availability of metabolic byproducts that contributed to reciprocity. To test this hypothesis, we performed a small subset of "hybrid" in silico experiments, where we analyzed the interactions that arose from one species being grown with oxygen and the other without oxygen. We studied the examples of *E. coli* with *B. subtilis* and *S. enterica*, whose pairwise simulations showed greater amounts of mutualistic interactions without oxygen (Fig. 6). These hybrid simulations demonstrated how an organism grown anoxically can provide a higher number of useful byproducts to its aerobic partner, leading to bidirectional interactions when both are grown without oxygen.

We also analyzed the interaction type distributions of our habitat-specific simulation sets. Though direct comparison between oxic and anoxic conditions was not possible with these organisms, we found their interaction patterns to be largely comparable to those in our core set. This was particularly evident when comparing competition and exchange patterns between our core set grown with oxygen and the aerobic aquatic and soil organisms (Supplementary Fig. 7e, f, h, i). We nonetheless noticed a substantial difference between the interactions predicted in the gut-associated microbes and our 14 core organisms grown anoxically. In simulations of gut-associated organisms, we predicted a lower frequency of competition and mutualism (Supplementary Fig. 7g, j). We suspect that the widespread costless secretion of amino acids by *Bifidobacterium adolescentis* and *Faecalibacterium prausnitzii* may be skewing these distributions, as an abundance of valuable secreted byproducts may preclude their partner organism from competing for primary carbon sources and equally contributing to an exchange.

**Costless secretions can produce stable interaction motifs.** Lastly, we combined data generated by our algorithm with ecological network simulations to understand how the simultaneous competition for common nutrients and cooperation through costless metabolite exchange could jointly affect the stability of pairwise consortia. Using the general interaction types outlined

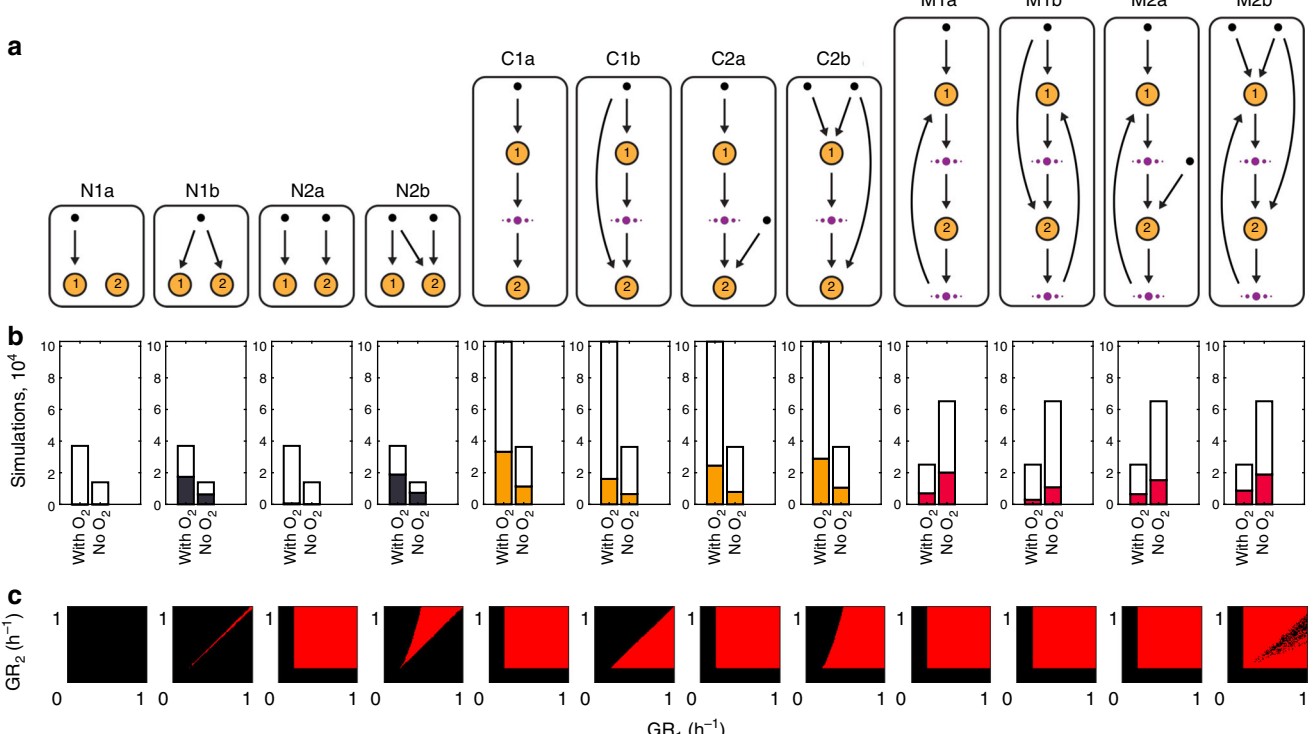

**Fig. 7** Interaction motif analysis and dynamical modeling of motif stability. **a** Schematic representation of specific motif types. Motifs are named according to three features: the interaction type (non-interacting, N; commensal, C; mutualistic, M), the number of carbon sources consumed by the pair (1–2), and competition for a primary carbon source (no competition, a; competition, b). Orange circles denote organisms, black dots denote primary carbon sources, and violet dots indicate any arbitrary number of costlessly shared metabolites. Arrows indicate direction of metabolite flow. **b** Frequency of specific motif types. Height of empty white bars indicate the total number of simulations that exhibited the general motif type (non-interacting, commensal, mutualistic). Colored bars within indicate the number of the specific motif type (N1a, N1b, and so on). **c** Stability space of motifs from dynamical chemostat modeling, as a function of the specific growth rates of the two organisms involved (GR$_1$, GR$_2$). Red indicates area of stable coculture

previously (non-interaction, commensalism, and mutualism with and without competition), we first enumerated all possible interaction network motifs (Fig. 7a) and calculated the frequency with which each motif was observed in our simulation set (Fig. 7b). For non-interacting motifs, our simulations predicted an almost exclusive representation of relationships involving competition for a primary carbon source. The distribution between competitive and non-competitive motifs was more balanced for commensal and mutualistic interactions, showing a slight preference for interactions involving competition.

In order to simulate how these interactions could contribute to stable symbioses, we created a dynamical chemostat model of two arbitrary species consuming carbon sources and exchanging costless metabolites according to each motif type (see Methods, Supplementary Fig. 9). By varying the maximum specific growth rates ($\mu_{max}$) of each species from 0 to 1 h$^{-1}$, we simulated the growth of the pair under each motif type for 500 h. If both species survived at the end of the simulation, we marked the motif type as stable at that combination of specific growth rates. We mapped the space of stable species pairs under each motif type, observing that competitive interactions generally had a reduced parameter space for enabling stability (Fig. 7c). Notably, though motif N1b was highly prevalent in the costless FBA simulation set, this motif represents competitive exclusion and cannot result in long-term stability. In contrast, though complete nutrient–organism orthogonality can yield stability over the whole space of parameters (N2a), this motif was not predicted to occur in the FBA simulations. An intermediate case between these two extremes (N2b) represents a balance between competition and independence with respect to external carbon source utilization: in this case, which

frequently occurs in our dataset, stability is achievable only for a narrow set of specific growth rates.

Our models predicted a marked increase in stability when costless metabolite exchange was enabled. In motif C1a, for example, the rate of costless metabolite secretion from organism 1 is enough to sustain organism 2, even when the specific growth rate of organism 2 is greater. This nonintuitive space of stable solutions is the result of the effective growth rate of organism 2 being reduced such that its rate of byproduct consumption does not exceed the rate of secretion by organism 1. Nonetheless, competition for primary carbon sources leads to decreases in the space of possible stable solutions, as observed in motifs C1b and C2b. In motif C2b, both organisms are competing for a carbon source and organism 1 is providing one or more costless metabolites to organism 2. Our dynamical modeling showed that the specific growth rate of organism 1 must usually be greater than that of organism 2 in order for both species to be stable. When feedback is allowed to occur (mutualism), the potential for stability greatly increases across our parameter space, even in the presence of competition for carbon sources. These motifs, with their associated prevalence data and dynamical properties, can overall serve as an atlas for guiding the engineering of stable synthetic consortia built off of costless metabolic relationships.

## Discussion

We have investigated the pairwise growth phenotypes and interactions of 24 diverse microbial species in over 2 million computational experiments. We found that resource-poor environments can provide the basis for the release of a wide variety of useful metabolic products secreted without cost by their producing

organism; these costless metabolic products provide, in a manner chiefly dependent on oxygen availability and environmental composition, valuable environmental enrichment, nearly doubling the potential of minimal environments to sustain growth. We further found that exchange of costless metabolites established beneficial uni- and bidirectional interspecies interactions, associated with different chances of stability of the ensuing consortia.

Our iterative medium expansion method allowed us to observe which metabolites were secreted in response to others in a mechanistic fashion, highlighting the capability of costless metabolites to enrich minimal environments and sustain biodiversity, even when organisms were competing for the same primary nutrients. Though spatial segregation of competing organisms can also allow for stable communities[9,39,54], purely competitive phenotypes can lead to community collapse in homogenous environments, which can limit taxonomic diversity in nature[55]. By studying the prevalence of mutually beneficial interactions in the presence of competition, our study can help understand metabolic dynamics in resource-poor environments, such as the oligotrophic communities found in the open ocean. Costless secretions, some of which resemble the metabolic leakages that underlie the Black Queen Hypothesis[18,21], may also contribute to the maintenance of small genomes in resource-poor environments, as the metabolic needs of some organisms can be fulfilled by others. Interestingly, the prediction that resource-poor environments can lead to diverse secretions and thus to the emergence and maintenance of beneficial interspecies interactions is consistent with prior suggestions that resource abundance or lack of stress may reduce the reliance of communities on metabolic exchange[56-60].

Our interaction analysis provides deeper mechanistic insight into the increased prevalence of mutualistic interactions without oxygen, a phenomenon that has been previously predicted computationally[27] and that provides a window into metabolic relationships in environments harboring steep oxygen gradients, such as the human gut[61]. By carrying out a set of hybrid oxic–anoxic in silico experiments, we observed that the additional metabolites secreted anoxically by a facultative anaerobe could provide valuable nutrients for aerobically growing organisms. This phenomenon has been suggested to play an important role in maintaining equilibrium in communities at oxic–anoxic interfaces in the mammalian gut[62,63] and could be the subject of further mechanistic studies.

Although our modeling method considered a wide space of mechanistic constraints in predicting costless metabolic exchange, we acknowledge that secretion patterns and exchange potential are also defined by a variety of other biological factors that fall outside the scope of our modeling framework[29], such as temperature[30], signaling and regulatory-based decisions[31-35], pH changes and metabolite toxicity[36], and concentration-dependent thermodynamic gradients[30,64]. In addition, the cost of metabolite secretion and the cooperative or competitive nature of an interaction may change when interpreted across different timescales[23], a quality that is not fully captured by the framework described here. In spite of these limitations, however, our analysis is able to demonstrate the plausibility of widespread costless cross-feeding in nature. Our results can also serve as a basis for prioritizing future specific experiments, for which model predictions could be thought of as a null hypothesis against which to compare empirical measurements. Dynamical modeling coupled with these metabolic analyses could then be used to obtain the parameter space most likely to yield desired stable partnerships in vivo. Because this approach relies on screening, in a scalable way, synergy-inducing environments, as opposed to engineering individual strains, it can simplify the process of assembling synthetic communities[65] and enhance our understanding of microbiomes.

## Methods

**Selection and modification of genome-scale metabolic models**. A genome-scale metabolic reconstruction was obtained for each of the 14 facultative anaerobic organisms used in the analysis, as well as for the organisms used in our habitat-specific simulations (Supplementary Data 1). Genome-scale metabolic models are mathematical representations of an organism's known metabolic network, which are used to generate mechanistic predictions of growth and resource allocation in a variety of environmental conditions. The process of generating a genome-scale metabolic model has been outlined conceptually[66-69] and described procedurally[70] by various groups, and generally comprises an automatic generation of a model based on pathway and genome data followed by manual curation by integrating phenotyping, metabolomic, or transcriptomic data[71]. We note that although an automatically generated draft metabolic model can be constructed for virtually any organism for which a genome annotation exists, the space of high-quality, experimentally verified metabolic models that have undergone the manual curation process summarized above is comparatively very small[72]. This is due to the time and resources needed to complete the curation process, which can span from 6 months[70] to >10 years for the iteratively refined model of *E. coli* K-12[73]. We nonetheless consider this process to be essential in producing models that can generate the mechanistic cross-feeding predictions detailed here, which rely on verified metabolic capabilities in monoculture.

The models used in this analysis span five taxonomic phyla, as well as a variety of primary metabolic strategies (Supplementary Data 1). In addition, these models describe several organisms that are commonly used for in vivo studies (*E. coli* K-12, *S. enterica* LT2, and so on), making the resulting costless cross-feeding predictions particularly useful for synthetic ecology experiments and microbial community assembly. Importantly, each metabolic model includes reactions that account for the energy requirements of organism growth, as well as those of metabolite production and secretion. These requirements are often incorporated as two key reactions: (1) NGAM, or non-growth-associated ATP maintenance, which comprises an ATP hydrolysis step that simulates ATP usage for processes that are not needed for growth; and (2) GAM, or growth-associated ATP maintenance, which accounts for energy usage in growth-associated processes such as macromolecule and protein synthesis. A minimum level of flux (typically determined experimentally) must flow through each of these reactions in order for the models to grow in silico. In this way, energetic costs of growth and metabolite production and transport are accounted for in the models.

Each model was imported into MATLAB (The MathWorks, Inc., Natick, MA) using the constraint-based reconstruction and analysis (COBRA) Toolbox[74], a software platform for constraint-based modeling of metabolic networks. In order to enable in silico cross-feeding to be correctly classified, the namespace of all of the metabolic compounds in each of the models was standardized to be internally consistent. This was performed via a computational pipeline with additional manual curation for irregularly annotated metabolites.

**Computational methodology description and inputs**. Our computational method comprises a set of programs written in MATLAB that use FBA to mechanistically define the growth status and metabolic exchange of microbes through costlessly secreted byproducts. Briefly, FBA is a mathematical method that determines an optimal distribution of metabolic flux through a biochemical network that will maximize a given objective, usually biomass[24,70]. An FBA problem is framed in the context of several constraints, namely: (i) $S$, the stoichiometric matrix of dimensions $m \times n$ where $m$ is the number of metabolites and $n$ is the number of reactions in the model; (ii) $\mathbf{v}$, the vector of all reaction fluxes; and (iii) $v_{min}$ and $v_{max}$, flux constraints placed on $\mathbf{v}$, defined by enzymatic capacity and experimentally measured uptake rates.

We employ FBA to determine if an organism is able to grow on the in silico growth media conditions we define, in addition to which metabolites are taken up and costlessly secreted. We first apply FBA by maximizing for growth and obtaining an optimal growth rate for an organism, $v_{g,0}$. To determine which metabolites are secreted costlessly, we set this growth rate as a minimum for the biomass flux and apply FBA again, recording any metabolites that were secreted and the new growth rate, $v_{g,s}$. We also apply the additional constraint of minimizing all reaction fluxes across the network to more closely simulate efficient use of the proteome and minimize cycling of metabolites through the network[75]. Our linear program therefore becomes:

$$\min|\mathbf{v}|$$

s.t.:

$$S \times \mathbf{v} = 0,$$

$$v_{min} \leq \mathbf{v} \leq v_{max},$$

$$v_{g,s} \geq v_{g,0}.$$

This optimization aims to encompass any enzymatic cost incurred by the organism in synthesizing and exporting any metabolite we deem to be costless. During each step in which growth or metabolite absorption and secretion are computed, FBA optimizations are performed separately for each in silico organism $i$ and $j$, with biomass production set as the objective function while minimizing the sum of the absolute value of **v**. Because we focus on the emergence of potential metabolic exchange through the availability of costlessly secreted metabolites, our modeling framework purposefully keeps FBA optimizations separate for each model without accounting for spatial or temporal community structure. It is also for this reason that we establish the biomass fluxes of each in silico organism as the objective functions to be optimized, as we are concerned with secretion of potentially useful metabolic byproducts that arise out of "selfish" optimal growth. This assumption of maximum growth with proteome optimality is also key for translating these organisms and predictions to in vivo synthetic ecologies, where biomass optimization more closely describes the behavior of organisms in batch or continuous culture[76]. In simulations where we employed alternative objective functions, we optimized either minimization of biomass production, maximization of ATP production, or minimization of ATP production (all with a lower bound for biomass flux set at 0.01 per hour) to compare secretion profiles to those observed under biomass maximization. For ATP maximization and minimization functions, we set the ATP maintenance reaction in each of the models as the objective.

Our algorithm requires six inputs: (1) a data structure containing the genome-scale metabolic models to be used, (2) a list of carbon sources, (3) the number $N_M$ of in silico organisms to be simulated together (for pairwise simulations $N_M = 2$), (4) the number $N_{CS}$ of carbon sources to be provided to each simulation, (5) a Boolean variable $\Omega = \{1,0\}$ that specifies if oxygen will be made available to the in silico organisms, and (6) a list of metabolites that makes up a simulated base growth medium, $M_{min}$. This base medium contains various nitrogen, sulfur, and phosphorus sources, as well as vitamins, ions, and metals needed for growth of the organisms (Supplementary Table 2).

We focused on pairwise species growth with two carbon sources ($N_M, N_{CS} = 2$). Although each genome-scale metabolic model we used has been manually curated to reflect in vivo metabolic capabilities, very few experiments have been performed to verify FBA-generated predictions for more than a single species[39,77]. We therefore limit the number of in silico species to two, in order to interpret the growth and cross-feeding predictions with greater confidence. This limit also constrains the combinatorial space of the simulations, which grows exponentially and becomes numerically intractable with more models and carbon sources. In addition, limiting simulations to $N_M = 2$ allows for greater experimental accessibility for assembling synthetic ecologies based on costless metabolite exchange. Our algorithm can nonetheless be applied to any $\{N_M, N_{CS} > 0\}$.

The list of all possible carbon sources was defined primarily from the carbon sources contained in the BIOLOG Phenotyping MicroArray 1 (PM1) plate, which is used for phenotyping and curation of genome-scale metabolic models[78–80]. The carbon sources we selected are common mono-, di-, and polysaccharides, all 20 amino acids, dipeptides, and organic acids contained in the PM1 plate. We also supplemented the list with additional carbon sources known to be consumed by the in silico organisms, for a total of 108 (Supplementary Table 1).

To permit uptake of the metabolites in the medium, the constraint on the uptake flux bound $v_{max}$ for each exchange reaction pertaining to a medium metabolite was removed in each of the models $i$ and $j$. This bound was fully removed ($v_{max} = 1000 \text{ mmol} \times \text{gDW}^{-1} \times \text{h}^{-1}$) for non-limiting medium components, and was set to ($v_{max} = 10 \text{ mmol} \times \text{gDW}^{-1} \times \text{h}^{-1}$) for the growth-limiting carbon sources $\alpha$ and $\beta$. This latter value is drawn from experimentally estimated uptake rates of sugars by $E.\ coli$ in exponential growth conditions[73], and is applied equally to all other species to simulate general availability of the carbon sources in the environment. In all aerobic simulations, the maximum uptake rate for oxygen was also unconstrained ($v_{max} = 1000 \text{ mmol} \times \text{gDW}^{-1} \times \text{h}^{-1}$), so as not to explicitly model conditions similar to those of overflow metabolism. All other exchange reaction $v_{max}$ values are set to zero to block uptake of metabolites not in the medium.

**Computing growth, secretion, and cross-feeding**. We describe the FBA operations at the core of our algorithm as a function $F$ that, given a medium condition **M** and organisms $i$ and $j$, outputs the binary growth status **g** of the organisms, as well as the set of metabolites **σ** secreted costlessly by the organisms:

$$F(\{\mathbf{M}, i, j\}) = \{\mathbf{g}, \boldsymbol{\sigma}\}. \tag{2}$$

Each in silico experiment $E$ for a given organism pair with a pair of carbon sources is made up of an initialization step, an expansion step consisting of series of applications of $F$, and a completion step (Supplementary Fig. 2). In the initialization step, two organisms $i$ and $j$ are selected, and a medium $\mathbf{M}_0$ is defined. $\mathbf{M}_0$ contains the minimal medium $\mathbf{M}_{min}$, two carbon sources $\alpha$ and $\beta$, and the variable $\Omega$, which denotes the presence or absence of oxygen.

In the expansion step, the function $F$ is applied for a series of iterations $c$. In each iteration, $F$ simulates the growth of both organisms in the current medium condition and returns the Boolean growth statuses $\mathbf{g}_c = \{g_i g_j\}$ (where $g_i g_j = \{0,1\}$) of both organisms and the set of any costlessly secreted metabolites, $\sigma_c$. To avoid recording metabolites reported to be secreted only as a result of numerical

uncertainty in FBA, a minimal lower flux bound of $0.01 \text{ mmol} \times \text{gDW}^{-1} \times \text{h}^{-1}$ was applied as a cutoff for determining secretion. If at least one organism in the pair grows, the medium is supplemented with $\boldsymbol{\sigma}_c$:

$$\mathbf{M}_{c+1} = \mathbf{M}_c + \boldsymbol{\sigma}_c. \tag{3}$$

As long as new metabolites continue to be secreted into the medium, that is,

$$\mathbf{M}_c > \mathbf{M}_{c-1}, \tag{4}$$

$F$ continues to be applied. This stepwise expansion simulates the organisms responding to the costlessly secreted metabolites being secreted and generating a richer medium. The completion step occurs when no new metabolites are secreted,

$$\mathbf{M}_c == \mathbf{M}_{c-1} \tag{5}$$

and the final iteration before this stabilization occurs is defined as $c_S$. Our algorithm therefore carries out individual in silico experiments $E_{i,j}^{\alpha,\beta,\Omega}$, defined as the output resulting from $c_s$ applications of $F$ given organisms $i$ and $j$, carbon sources $\alpha$ and $\beta$, and the presence or absence $\Omega$ of oxygen:

$$E_{i,j}^{\alpha,\beta,\Omega} \equiv \left\{\mathbf{g}_c, \mathbf{M}_c\right\}_{c=1}^{c_s} = F(\{\mathbf{M}_0, i, j\})_{c=1}^{c_s}. \tag{6}$$

**Dynamical modeling of interaction motifs**. We designed a dynamical modeling method to simulate the long-term stability of each pairwise interaction type observed in our in silico experiments. We first established a graph theory framework to map each simulation to a specific interaction motif, each of which accounted for the general interaction type (non-interacting, commensal, or mutualistic), the number of carbon sources consumed by the pair, and the competition status for the carbon sources ("a" denotes no competition, "b" denotes competition) (Fig. 5a). We next applied a differential equation-based growth model to each specific motif. Since motifs with two carbon sources can be represented by more than one motif topology, we selected one representative topology from these motifs to simplify the space of dynamical modeling simulations. These equations were modeled off Monod dynamics[81] and are intended to simulate growth of species in a chemostat, with constant replenishment of medium components. The abundance of each organism $s_i$, in g/L, is modeled as follows:

$$\frac{ds_i}{dt} = s_i \mu_{max,i} \left(\frac{m_\alpha}{k_{s_i, m_\alpha} + m_\alpha}\right) - Ds_i, \tag{7}$$

where $\mu_{max,i}$ is the maximum specific growth rate of organism i in $\text{h}^{-1}$, $m_\alpha$ is the concentration of carbon source $\alpha$ in $\text{g} \times \text{L}^{-1}$, $k_{s_i, m_\alpha}$ is the concentration of $\alpha$ at which organism $i$ reaches half its maximal growth rate in $\text{g} \times \text{L}^{-1}$, and $D$ is the chemostat dilution rate in $\text{h}^{-1}$. If two carbon sources are present and the organism is determined to take up both by the motif definition, the equation is modified to include a carbon source $\beta$ as follows:

$$\frac{ds_i}{dt} = s_i \mu_{max,i} \left(\frac{m_\alpha}{k_{s_i, m_\alpha} + m_\alpha}\right) \left(\frac{m_\beta}{k_{s_i, m_\beta} + m_\beta}\right) - Ds_i. \tag{8}$$

The concentrations of each carbon source are defined as follows:

$$\frac{dm_\alpha}{dt} = I_{m_\alpha} - \frac{s_i}{K_{m_\alpha}} \mu_{max,i} \left(\frac{m_\alpha}{k_{s_i, m_\alpha} + m_\alpha}\right) - Dm_\alpha, \tag{9}$$

where $I_{m_\alpha}$ is the nutrient stock concentration for $m_\alpha$ in $\text{g} \times \text{L}^{-1}$, and $K_{m_\alpha}$ is the ratio of nutrient consumed by the organism i in $g_{nutrient} \times g_{cells}^{-1}$. This equation is modified with an additional term (organism $j$) to simulate competition for $m_\alpha$.

To simulate metabolic exchange, equations for the abundances of costlessly produced metabolites ($\tilde{m}$) in $\text{g} \times \text{L}^{-1}$ were defined as follows:

$$\frac{d\tilde{m}_i}{dt} = k_{\tilde{m}_i} \times s_i - \frac{s_j}{K_{\tilde{m}_i, s_j}} \mu_{max,j} \left(\frac{\tilde{m}_i}{k_{s_j, \tilde{m}_i} + \tilde{m}_i}\right) - D\tilde{m}_i \tag{10}$$

Here, metabolite $\tilde{m}_i$ is produced by organism $i$ and consumed by organism $j$. $k_{\tilde{m}_i}$ is the synthesis rate of the metabolite in $\text{h}^{-1}$, $K_{\tilde{m}_i, s_j}$ is the ratio of metabolite consumed by the population $s_j$ in $g_{metabolite} \times g_{cells}^{-1}$, and $k_{s_j, \tilde{m}_i}$ is the concentration of metabolite needed for the population $s_j$ to reach half of its maximum growth rate in $\text{g} \times \text{L}^{-1}$.

We then combine Eqs. (7, 9, and 10) to fit the particular motif being modeled (Supplementary Fig. 9). The values of the parameter values are described in Supplementary Table 3 and are based on values reported by Smith[82], Balagaddé et al.[83], and those based on reasonable estimates for resource consumption. For

each motif, we vary the maximum specific growth rate of both organisms from 0 to 1 per hour and run the simulation for 500 h. If both organism abundances are above 0.05 g × L$^{-1}$ at the end of the simulation, we determine the motif to be stable at the prescribed growth rates.

**Support vector machine classification.** We trained three separate support vector machine (SVM) classifiers to quantify the degree to which oxygen availability, species identity, and carbon source type contribute to the variability in secretion profiles. SVMs were constructed using the MATLAB function "fitcsvm" for the two-class oxygen availability vector, and using the MATLAB function "fitcecoc" for the multi-class species and carbon source category vectors. Secretion profiles were represented as a binary matrix, with each row representing a simulation and each column denoting a metabolite. Cross-validation was performed by: (1) randomly partitioning the matrix of secreted metabolites into 10 sets of equal size, (2) training the classifier on nine of the sets and testing on the remaining sets, (3) repeating training and testing for the remaining nine partitions, and (4) combining accuracy statistics for each set.

**Dynamic flux balance analysis simulations.** We carried out a set of simulations to determine the effect of substrate concentration on metabolite secretion patterns. For these simulations, we used the COMETS (computation of microbial ecosystems in time and space) software package[39], which uses dFBA[38] to integrate metabolic fluxes over sequential time intervals. This method enables specification of substrate concentrations as well as measurement of biomass growth and byproduct concentrations. As dFBA is more time- and computationally intensive than our primary FBA-based algorithm, we limited the number of COMETS simulations to analyzing single organisms growing on one carbon source. For each organism–carbon source pair, we ran COMETS with three different carbon source concentrations (0.01, 20, and 200 mM) for a simulated 2 h to capture differences in secreted metabolites. These combinations yielded a total of 9072 distinct simulations.

**Code availability.** Code for running pairwise cross-feeding simulations is available at github.com/segrelab/CostlessExchange.

## Data availability

All supplementary tables are contained in the Supplementary Information file. Additional data are provided in Supplementary Data files 1 and 2. Raw results data for all pairwise cross-feeding simulations are available at github.com/segrelab/CostlessExchange. A Reporting Summary for this Article is available as a Supplementary Information file.

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

## Acknowledgements

We thank Dr. Niels Klitgord for pioneering ideas that inspired launch of this work. We are also grateful to David Bernstein, Joshua E. Goldford, Meghan Thommes, Demetrius DiMucci, and all members of the Segrè Lab for helpful discussions. A.R.P. is supported by a National Academies of Sciences, Engineering, and Medicine Ford Foundation Pre-doctoral Fellowship and a Howard Hughes Medical Institute Gilliam Fellowship. This work was supported by funding from the Defense Advanced Research Projects Agency (purchase request no. HR0011515303, contract no. HR0011-15-C-0091), the U.S. Department of Energy (grants DE-SC0004962 and DE-SC0012627), the NIH (grants 5R01DE024468, R01GM121950, and Sub_P30DK036836_P&F), the National Science Foundation (grants 1457695 and NSFOCE-BSF 1635070), MURI Grant W911NF-12-1-0390, the Human Frontiers Science Program (grant RGP0020/2016), and the Boston University Inter-disciplinary Biomedical Research Office.

## Author contributions

A.R.P. and D.S. designed the research. A.R.P. designed the computational framework, carried out all simulations, and conducted data analysis. M.M. contributed to the generation of standardized genome-scale models. A.R.P. and D.S. wrote the manuscript. All authors read and approved the final manuscript.

## Additional information

**Competing interests:** The authors declare no competing interests.

