## [Peer Review File · Nature Communications]

Reviewers' Comments:

Reviewer #1:

Remarks to the Author:

This is a very well written manuscript that addresses timely questions in microbial ecology and metabolism. I very much appreciated the manuscript and I believe it could be of broad interest to the field. The manuscript elegantly investigates how the costless secretion of metabolites could emerge and enable the coexistence of different microbial genotypes. While the manuscript lacks experimentation to support any of the findings, the results are nevertheless exciting, and the authors made clear that the manuscript should be viewed as generating hypotheses rather than conclusively identifying costless secretions and metabolic interactions. While I enjoyed the manuscript and believe it could be of broad interest, I nevertheless have several critical comments.

GENERAL COMMENTS

1) The authors clearly acknowledge that the work is exploratory in nature, with one goal being to "produce a global atlas of expected, environment-dependent costless secretions (Lines 478-479)". While I agree that an atlas could help guide future experiments, there is no assessment of the false positive rate of the hypothesised secretions. What is the probability that a predicted costless secretion does indeed take place? What is the probability that the predicted costless secretion is indeed costless? My fear is that, if the false positive rate is very high, than generating such an "atlas" could be more detrimental than helpful to the community as a whole. For example, it could potentially catalyse unacceptably high-risk research efforts. I would highly encourage the authors to address or at least acknowledge this issue and, if necessary, provide sufficient warning to any potential user of such an "atlas" that the false positive rate remains unknown.

2) I cant find the supplementary information. It is not contained in the merged reviewer documents. I am interested in knowing the exact composition of the minimal medium, and am specifically wondering whether nitrate was provided. If nitrate was provided, how does this impact fermentation processes? At low substrate concentrations, I assume cells would preferably respire nitrate than ferment.

3) I am confused about the focus on "resource poor environments". For example, I am confused by the following statement: "We found that resource-poor environments provide the basis for release of a wide variety of useful metabolic products secreted without cost... (Lines 398-399)". Why would resource-poor environments in particular promote the costless secretion of metabolites? Naively, I would expect resource-rich environments to be more effective at promoting the release of such metabolites.

4) On that note, does the set of potential costless secretions depend on substrate concentration? This seems important to address.

5) One cost that may be associated with metabolite secretion is metabolite toxicity. If metabolites are secreted, they could potentially accumulate to toxic concentrations. Is metabolite toxicity accounted for when assessing whether the secretion of a particular metabolite is costly or not? This seems essential.

6) All of the costless metabolites are identified using the maximisation of growth as the objective function. Are the results robust if using a different objective function? From my perspective, narrowing the set of potential costless secretions to those that are robust under different objective functions

might be a way to improve confidence in the results.

7) "Costless" seems to have a time dependence to it. Metabolite secretion might have no effect on growth over a certain time-frame, and thus be costless over that time-frame. However, if the metabolite can be taken back up by the producer at a later time, does this mean that secretion does have a cost over a longer time frame? If so, how would this affect the main conclusions?

SPECIFIC COMMENTS

Lines 16-18: ".. we show how the exchange of costless metabolites can facilitate the engineering...". This study does not show how this could be done. It simply proposes a strategy to achieve this. Please rephrase.

Lines 26-28: What about temporal mechanisms that could explain the maintenance of diversity?

Lines 30-31: "or by maintaining thermodynamic gradients..." Is this example really susceptible to cheating? I typically think of this as costless byproduct cross-feeding.

Lines 39-42: Im not sure anyone thinks that metabolic exchange alone could account for the degree of biodiversity observed in nature. It is simply one of many mechanisms that could promote biodiversity. An open question is whether it plays a minor or predominant role in maintaining and promoting biodiversity. Restate?

Lines 44-45: "radically different mechanism". I think this is an overstatement. Research regarding costless byproduct cross-feeding predates research regarding costly cross-feeding. The recent shift towards investigating costly cross-feeding was largely to address social evolutionary questions, not to propose it as more or less prevalent than costless byproduct cross-feeding.

Reviewer #2:

Remarks to the Author:

The manuscript by Pacheco et al. uses metabolic models to chart "costless" metabolite secretions that can lead to inter-species interactions such as cooperation (beginning of which is still an open question in ecology). This question has previously been tackled, also using metabolic models, by several publications (including landmark papers from the same group). Compared to the previous studies, this work considers multiple nutrient contexts (highly relevant from ecological perspective), explicit cost on growth, and looks into different possible interaction motifs and their stability in a steady-state (simulated as chemostat) context. Overall, the manuscript provides a comprehensive perspective on possible metabolite exchanges in microbial communities. This comprehensive analysis is a major advance in understanding the broad role that metabolic interactions play in microbial communities. There are few comments (mostly semantics, missed discussion points etc.) that I recommend authors to consider for revising the manuscript.

-The use of "radically different" is not be justified in my opinion since the idea of costless secretion as beginning of cooperation is not new in ecology. Also see my next comment.

- The term costless is not accurate in the context of this study. The analogy of lion->vulture is not applicable in the case of metabolite secretion. Excretion of metabolites, even if they are overflowing,

do have a cost in terms of production as well as transport (both protein mediated or passive secretions take some resources; even if little, it can have fitness implications). These resource requirements are not accounted by the metabolic models (at least not by the kind used in this study). Secondly, overflow metabolism is often due to some benefit to the organism, e.g. rapid substrate depletion / fast growth (e.g. ethanol secretion by yeast – see e.g. Nilsson and Nielsen, 2016) or avoiding toxicity (e.g. N-overflow by yeast – see e.g. Ponomarova et al., 2017). This is why usually the cost is carried by the secretors.

- The term cooperation in ecology is often used to imply evolution for that specific purpose. Thus, a better term to use here is mutualism, unless and until the authors can show evidence for evolutionary selection.

- The secretion of AA /other metabolites can be highly regulatory context dependent – in case of yeast it requires abundant N availability and signal for NCR de-repression (e.g. poor-quality N source or perturbation of TOR pathway). These effects are not accounted for by the metabolic models used. This limitation should be discussed in the manuscript.

- The methods section does not specify what is the maximum oxygen uptake rate imposed under aerobic conditions. Usually there is no secretion of, e.g. the classic overflow metabolites in the model (acetate, ethanol etc.), unless one simulates overflow metabolism by artificially limiting the maximum oxygen uptake rate (or using molecular crowding or other resource allocation considerations). The authors should be clearer on the choices made here.

- Something seems wrong with Figure 7c (maybe the order of the plots is incorrect). I would expect growth-coupling of species 2 to species 1 in cases C1a and M1a. On the other hand, in M1b I would expect a symmetric plot since the topology is also symmetric.

- The simulated chemostat experiments are unclear. The (pseudo) steady-state growth rates in a chemostat need equal the dilution rate, so how two species can have different growth rates in a chemostat?

- Along the paper the authors often mention cases where predicted interactions match known interactions from the literature. These cases should be presented together into a supplementary table.

Reviewer #3:

Remarks to the Author:

The present study assesses the landscape of cross-feeding interactions that can arise in a generic set of facultative anaerobic microbial organisms across many environmental conditions. The authors use an exhaustive modelling approach based on full metabolic models of the species to determine the potential of metabolites to arise and be secreted costlessly and subsequently promote one-directional and bidirectional ecological interactions. The main hypothesis states that costless metabolites drive the emergence of microbial community interactions.

Simulations suggest that the cost of metabolite 'production' varies with environment, in particular with alternative organisms and produced metabolites being present in the microenvironment. The obtained results indicate oxygen availability as strong influencer of variety (decreasing) and number (increasing) of costless metabolites generated and the number of potential mutualistic (decreasing) interactions. Furthermore they find that oxygen availability also drives patterns of co-occurring metabolite production that overall enhance growth capabilities for the present organisms. The authors

link their results with ecological theory for explaining stable biodiversity and show that synergistic mutualistic interactions (i.e. positive feedbacks) are the main driver of system stability, even if the organisms compete for resources at the same time.

The study is well designed, uses cutting edge modelling and presents a very thorough interpretation of the results with the important objective to contribute novel insights to microbial community ecology. Understanding the drivers of community robustness and biodiversity in microbial communities (MCs) is of highest importance for basic research as well as for applications in public health, biotech or climate change research. The simulations also demonstrate clearly how the cost of metabolite production is a function of the environment and hence robustness of metabolic interactions based on costless production should also be.

There are however some shortcomings that are of technical nature that limit the impact of the study. First and foremost, the choice of organisms is based on availability of curated metabolic models as discussed in the text. This is understandable from a technical point of view, but inevitably raises the question whether studying the combinatorics of potential metabolic overlaps is meaningful for real scenarios. More profoundly, the question arises whether real communities would show distinct patterns due to adaptive processes within the habitat. As no additional material is presented that can address this aspect (e.g. case study for particular habitat, clear binning of organisms in the used data set which likely live together or which have exclusive habitats), the main argument is ultimately not conclusive. Based on the evidence, it cannot be decided whether observed patterns either represent the natural (random) background or some fundamental rules of microbiome evolution. The expectations raised by the exciting study setup are unfortunately missed and sold below value.

A second shortcoming, which is also pointed out by the authors, is the lack of including microbial pH requirements into the model. The model simulates the production of metabolites that are thereafter available in the medium and can serve to support biomass production of interacting organisms. As medium composition is known at each point in the simulation, potential pH changes should be straightforward to approximate. These would have the advantage of contributing realistic boundary conditions for individual growth dynamics and these would reduce the otherwise unconstrained interaction space.

To exclude that variation in metabolite production is driven by oxygen availability or by the organisms the authors present a PCA analysis. Unfortunately the figure is over-plotted, too small and lacking important information with regards to the first principle components composition (axes!) that would allow to evaluate this conclusion. Indeed, a direct strategy to assess effect sizes of potential drivers (oxygen, species, interactions) would much better support this statement. For example principal component regression or MANOVA provide objective, comparable measures that enable ranking and conclusion.

Finally, the authors miss to set their study in context with the scientific field, including work published by Elhanan Borenstein, Shiri Freilich, Corina Tarnita or broader, symbiosis research.

Minor comments:

-One of the main outcomes is that mutualistic, positive interactions promote stable coexistence, compared to unidirectional commensalism – in other words increasing positive dependence enables stability. I find this is a very interesting theoretical result that should be given more focus in the presentation. It is in line with earlier results from the lab, yet should be discussed with respect to the idea that negative interactions contribute robustness (in spatial explicit systems).

-From the introduction I get the impression that the authors make a categorical distinction between selfish, unwanted and other interactions. I recommend modifying the section in accordance with current evolutionary theory.

-The central definition of costless production of metabolites requires a clearer explanation and I suggest to present the underlying math in the main text to support the definition.

-In several figure captions the text is still missing (fig3, fig 6, fig S6, fig S7, fig S8). Supplement 4 is referenced in the manuscript, but not available.

-Figure 1: Oxygen logos are too small to read when printed

-L145 typo: a *positive* shift from *oxic to anoxic*

-L163: presenting information on the individual metabolite shifts would be very valuable

-L293: How is this measure of cooperative potential distinct and advantageous from other measures used (e.g. Freilich et al 2011 Nat Comm 589)?

-L413: Emergent properties are considered higher order phenomena that are driven by low-level interactions.

We are grateful to the Editor and the Reviewers for evaluating our manuscript. We were glad to see the overall positive assessment of our work and very much appreciated the insightful questions and constructive comments raised. As detailed in our response letter below, we have addressed each point to the best of our abilities. We feel that our manuscript has significantly improved as a result of this process, both in terms of the clarity of the conclusions and of its place within the context of existing literature.

Reviewer #1 (Remarks to the Author):

This is a very well written manuscript that addresses timely questions in microbial ecology and metabolism. I very much appreciated the manuscript and I believe it could be of broad interest to the field. The manuscript elegantly investigates how the costless secretion of metabolites could emerge and enable the coexistence of different microbial genotypes. While the manuscript lacks experimentation to support any of the findings, the results are nevertheless exciting, and the authors made clear that the manuscript should be viewed as generating hypotheses rather than conclusively identifying costless secretions and metabolic interactions. While I enjoyed the manuscript and believe it could be of broad interest, I nevertheless have several critical comments.

We thank the reviewer for the positive evaluation of our work and for the important and constructive comments.

GENERAL COMMENTS

1. The authors clearly acknowledge that the work is exploratory in nature, with one goal being to "produce a global atlas of expected, environment-dependent costless secretions (Lines 478-479)". While I agree that an atlas could help guide future experiments, there is no assessment of the false positive rate of the hypothesised secretions. What is the probability that a predicted costless secretion does indeed take place? What is the probability that the predicted costless secretion is indeed costless? My fear is that, if the false positive rate is very high, than generating such an "atlas" could be more detrimental than helpful to the community as a whole. For example, it could potentially catalyse unacceptably high-risk research efforts. I would highly encourage the authors to address or at least acknowledge this issue and, if necessary, provide sufficient warning to any potential user of such an "atlas" that the false positive rate remains unknown.

We thank the reviewer for highlighting this very important point. In an effort to ensure accuracy of the reported secretion patterns, we have based our analysis on genome-scale metabolic models that have associated experimental validation data. This step is essential for making mechanistic predictions to the degree that we have done, as these models have been shown to reflect experimentally-observed metabolic activity with high fidelity. To more clearly detail this rationale, we have added sentences in the Results (line 214) and the Methods (line 628) sections commenting on the general accuracy of these models. We nonetheless agree with the reviewer that over-reporting secretion patterns may be detrimental to future experimental efforts, and have added a section in the discussion more explicitly acknowledging the limitations of our method (line 582).

Though it is enormously difficult to experimentally validate all of the secretion patterns predicted in our dataset, we have identified a number of studies that have observed a subset of our predictions experimentally. We have added references to these studies in Supplementary Information 5.

Moreover, we have edited the manuscript to more prominently discuss the important measures that we took to ensure the rate of false positive secretions was minimized. These measures, which supplement our choice to only use experimentally-curated genome-scale models, entail the application of following constraints: (1) Our application of FBA relies on minimizing the absolute value of the sum of all the fluxes in the network. Application of this constraint curtails superfluous network-wide metabolic flux that may result in secretion of further metabolites than realistically feasible (Methods line 667 and Results line 120) (2) We apply a small threshold that a metabolite's export rate must surpass in order to be reported as having been secreted. Using FBA, especially when simulating resource-poor environments, can result in reporting of near-zero flux rates for metabolite uptake and secretion due to numerical error. Our threshold eliminates reporting of such metabolite secretions, and we clarify the process in the text (line 124) and Methods (line 757).

2. I cant find the supplementary information. It is not contained in the merged reviewer documents. I am interested in knowing the exact composition of the minimal medium, and am specifically wondering whether nitrate was provided. If nitrate was provided, how does this impact fermentation processes? At low substrate concentrations, I assume cells would preferably respire nitrate than ferment.

We apologize for the difficulty in locating the supplementary information. We will of course make sure to upload all the revised material and label it appropriately, and we assume that the Editorial Staff would be able to assist in case any file seems difficult to locate. The supplementary figures are being uploaded as a single separate file, and the supplementary information is included as an Excel spreadsheet.

We wish to clarify that the minimal medium did include nitrate, which was fully reduced to nitrogen gas by at least one organism in 28,229 out of 286,484 anaerobic simulations in which at least one organism grew, suggesting that anaerobic respiration via nitrate did take place in a subset of the results. As our implementation of FBA does not explicitly define nutrient concentrations, it is difficult to quantify the effect of nitrate concentration on fermentation. Nonetheless, in the simulations where nitrate was taken up by an organism, we did observe a slight reduction in the number of secreted fermentation products (2.81 ± 1.11 metabolites for non-nitrate respirers vs 2.38 ± 0.54 metabolites for nitrate respirers). We have added a more detailed discussion of this observation in the text (line 198).

3. I am confused about the focus on "resource poor environments". For example, I am confused by the following statement: "We found that resource-poor environments provide the basis for release of a wide variety of useful metabolic products secreted without cost... (Lines 398-399)". Why would resource-poor environments in particular

promote the costless secretion of metabolites? Naively, I would expect resource-rich environments to be more effective at promoting the release of such metabolites.

We thank the reviewer for raising this question. In emphasizing the resource-poor nature of the environments used in our simulations, we wished to present the idea that resource abundance does not appear to be a necessary precursor for the release of costless metabolic products as would intuitively be assumed. Based on these secretion patterns, it was our intention to highlight that even environments that have minimal resources can foster the emergence of cooperative interactions in a wide variety of cases. In fact, it has been observed in a number of studies (PMID: 12075350, PMID: 23091010, PMID: 27557335, PMID: 19127304) that low resource abundance or the introduction of metabolically stressful conditions can in fact promote cooperative phenotypes in microbial and multicellular organisms. We have clarified our phrasing and have added a discussion on this effect in the text (line 544).

4. On that note, does the set of potential costless secretions depend on substrate concentration? This seems important to address.

We agree with the reviewer that this is an important point to address. As the FBA algorithm at the core of our framework does not consider substrate concentration (instead relying on maximum flux bounds to constrain each reaction), we have carried out a new series of simulations using the COMETS (Computation of Microbial Ecosystems in Time and Space) software package developed by our group (PMID 24794435). COMETS relies on dynamic flux balance analysis (dFBA), which integrates the growth and metabolite uptake and secretion fluxes of an organism over a specified time interval. This integration allows for concentrations of secreted metabolites to be predicted, as well as for concentrations of substrates to be specified. However, COMETS and dFBA implementations in general are much more resource- and time-intensive than the FBA algorithm at the core of our framework, making it difficult to scale such an analysis to our entire dataset.

To account for these resource limitations, we elected to run COMETS monoculture simulations for all organisms with a single carbon source, with and without oxygen. For each organism-carbon source pair, we introduced the carbon source at three increasing concentrations: 0.01 mM, 20 mM, and 200 mM (total of 9,072 simulations). We ran each scenario for a simulated two hours in order to obtain the identities of any secreted metabolites as the organisms grew. This simulation setup allowed us to directly examine whether or not the set of costless secretions depend on substrate concentration (Methods line 840).

We found that the set of costless secretions was the same for the vast majority of simulations (97.9% with oxygen, 98.0% without oxygen) regardless of substrate concentration. We report this result in the text (line 182) and list the metabolites in Supplementary Information 7. All organisms were involved in at least one of the few cases where the number of secreted metabolites differed between concentrations, though the vast majority of these cases featured a difference of only a single metabolite.

Given the small set of scenarios in which the metabolites differed with concentration, we were not able to discern any patterns between these scenarios and the carbon sources used as substrates.

This result may be due to a fundamental limitation of FBA-based modeling techniques, which largely rely on the presence of a metabolite (as opposed to its abundance) to determine whether or not it is favorable to take that metabolite up. Since the abundances of substrates were all nonzero in the scenarios we tested, any byproducts that were secreted as a result of metabolite uptake should be the same.

It may nonetheless be possible to capture some concentration-dependent substrate uptake and byproduct secretion patterns with dFBA. However, this would entail a much larger systematic study involving pairs of organisms, a wider range of substrate concentrations, as well as organism models with varying metabolite uptake capabilities. We regard such an undertaking to be outside the scope of this current study, primarily due to the difficulties in scaling dFBA to such a combinatorially large space. Nonetheless, our current dFBA analysis suggests low sensitivity of costless secretions to the range of substrate concentrations that we tested for individual microbes.

We however do not wish to imply that varying substrate concentrations will not have effects on longer timescales, especially as nutrients become depleted. It is intuitive that, if nutrient depletion does not cease growth, organisms must change their metabolic strategies to cope with such environmental changes. These changes in metabolism may then lead to altered metabolite secretion profiles and, consequently, cross-feeding dynamics that are different from those predicted when nutrients were available. We have added a mention of this question in the Discussion (line 582), though we feel that such longer-term consequences fall outside the scope of the questions we ask in this work, as we are concerned chiefly with the potential for secretions and interactions to emerge costlessly in a given environment.

5. One cost that may be associated with metabolite secretion is metabolite toxicity. If metabolites are secreted, they could potentially accumulate to toxic concentrations. Is metabolite toxicity accounted for when assessing whether the secretion of a particular metabolite is costly or not? This seems essential.

We thank the reviewer for raising this important point, as toxic metabolite accumulation can indeed severely impact organism growth and therefore metabolic exchange. A first step to begin addressing the impact of toxic metabolites is expanding our application of dFBA to all of our simulations and allowing them to run for longer simulated timescales (e.g. days). This analysis would allow us to predict the concentrations of all metabolites at all points of a cross-feeding simulation. However, we are still faced with two key limitations to further analysis of toxicity using constraint-based modeling. First, it is not immediately obvious what constitutes a toxic metabolite, as toxicity depends on a number of parameters in addition to concentration and target organism, such as environmental context and potential remediation by another organism. It is possible to predict remediation by a partner strain using dFBA, but the toxicity of a metabolite given

a particular environmental context would need to be determined experimentally for all the organisms, toxic metabolite concentrations, and environmental compositions we simulate: a study that remains experimentally intractable. Secondly, a key limitation of current FBA methods is that they are not equipped to consider the effects of toxic metabolites on organisms as they do not consider intracellular metabolite concentrations. dFBA simulates cell death with an explicit death rate, but to our knowledge has yet to be applied in a contextual manner with respect to metabolite concentrations. Thermodynamic FBA (PMID: 17172310) may be able to simulate the slowing down of metabolism due to saturating external metabolite concentrations, thereby capturing one potential cause of metabolite toxicity. We believe that incorporating this method would require fundamental restructuring of our study, we have added a mention of toxic metabolites to the Discussion (line 582), in order to acknowledge the limitations of our study and stimulate further study on this important topic.

6. All of the costless metabolites are identified using the maximisation of growth as the objective function. Are the results robust if using a different objective function? From my perspective, narrowing the set of potential costless secretions to those that are robust under different objective functions might be a way to improve confidence in the results.

Our decision to select maximization of growth as the main objective function derived from our desire to most closely simulate secretion of byproducts by organisms growing “selfishly,” and presumably as rapidly as possible. We nonetheless thank the reviewer for raising this important point and have carried out three new analyses with alternative objective functions to compare metabolite secretion profiles between all conditions. We carried out an additional ~3 million simulations with all organisms and carbon sources under aerobic and anaerobic conditions, with minimization of growth, maximization of ATP production, and minimization of ATP production as objective functions for the organisms. We term these alternative objective functions as minGro, maxATP, and minATP respectively, and term maximization of growth as maxGro.

*We found the metabolite secretion profiles in all three of these conditions to be very similar to those under maxGro. The most similar condition was maxATP, with only one metabolite (5'-Deoxyadenosine) being reported under maxGro and not under maxATP (3,148 simulations in maxGRO vs 0 in maxATP, all by *S. cerevisiae* without O₂). The conditions minGro and minATP were also virtually identical to each other, but the greatest difference in secretion profiles was observed between the maximization and minimization conditions. Nonetheless, there were only 10 metabolites reported under maxGro that were not present in minGro or minATP and the total number of simulations in which these metabolites were over-reported in maxGro account for 0.18% of all predicted metabolic secretions. We have reported these secretions in Supplementary Information 6.*

Using the number of simulations in which a metabolite was secreted (N_S), we found that the differences in N_S were centered around zero ($\mu = 0.002 \pm 0.033$, normalized to N_S under maxGro), with no organism having a difference of more than 0.20 relative to the

value of N_S under maxGro. When comparing between conditions for all organisms, we found that values of N_S were highly correlated between maxGro and minGro ($R^2=0.95$), maxATP ($R^2=0.99$), and minATP ($R^2=0.95$).

We have reported these results in the text (line 222) and have outlined our methodology in the Methods (line 693). We have also compiled these results into a new supplementary figure (Figure S5).

7. "Costless" seems to have a time dependence to it. Metabolite secretion might have no effect on growth over a certain time-frame, and thus be costless over that time-frame. However, if the metabolite can be taken back up by the producer at a later time, does this mean that secretion does have a cost over a longer time frame? If so, how would this affect the main conclusions?

Our simulations consider the possibility of a producing organism taking a metabolite back up after secreting it, as all costless metabolites are made available to both organisms upon secretion. Since the growth rate and metabolic fluxes of the organisms are calculated every time the environment changes, the cost of secreting a costless metabolite is always zero within each new context. We may therefore say the "instantaneous" cost of costless metabolite secretion is always zero in our framework.

Moreover, since our algorithm continues to update the environment until no new metabolites are secreted, the secretion profiles that we obtain at the end of a simulation represents the set of metabolites produced without cost once the system equilibrates. Thus, the final set of metabolites may be thought of as the result of integrating the previous instantaneous and costless metabolites.

Nonetheless, this may not fully address the very interesting question posed by the reviewer, as instantaneous cost may differ from long term cost. If a costlessly-secreted metabolite were to be taken back up after a certain amount of time, it may indicate that the organism can no longer survive on the primary substrate and must switch to using its own byproducts for growth. One well-known example of this scenario is the "acetate switch," which describes the phenomenon in which E. coli begins to scavenge for its previously-secreted acetate after depleting other carbon sources (PMID: 15755952). It is difficult to determine whether or not acetate secretion in this entire scenario is costless. Our framework would lead us to answer in the affirmative, as the initial secretion of acetate by E. coli was within the context of free byproduct secretion. It is not clear if the later dependence on this waste metabolite for growth would negate its null metabolic cost.

A similar phenomenon has been predicted computationally using methods similar to ours (PMID: 22638572). Here, Beg et al. report that a genome-scale model of Shewanella oneidensis secreted and later consumed pyruvate when lactate was used as a carbon source. The authors hypothesized that this may have been due to the rate of intracellular catabolism of lactate into pyruvate outpacing the cell's capability to use pyruvate, thus necessitating its secretion. Under this hypothesis, we may infer that the

secretion of pyruvate would also be costless, as accumulating pyruvate in the cell may lead to slowing of metabolic activity and growth. This scenario mirrors panel c in our Figure S1.

There may also exist scenarios in nature where a costly metabolite (e.g. an amino acid) is secreted, re-absorbed, and then later secreted again without cost. Our simulations do not currently take into account these nuanced scenarios, which to us represent an exciting area for future analysis and refinement. In this manuscript, we have added a sentence in the Discussion that addresses this question (line 582).

SPECIFIC COMMENTS

8. Lines 16-18: ".. we show how the exchange of costless metabolites can facilitate the engineering...". This study does not show how this could be done. It simply proposes a strategy to achieve this. Please rephrase.

The sentence has been rephrased (line 14).

9. Lines 26-28: What about temporal mechanisms that could explain the maintenance of diversity?

We have added references (line 25) that also show how temporal dynamics can stabilize microbial communities.

10. Lines 30-31: "or by maintaining thermodynamic gradients..." Is this example really susceptible to cheating? I typically think of this as costless byproduct cross-feeding.

We thank the reviewer for raising this point. We agree that it may not have been an appropriate example to contrast with costless byproduct cross-feeding, and have edited the sentence in question accordingly (line 31). Nonetheless, we feel the paper referenced marks an important milestone in the development of genome-scale metabolic modeling, so we have included it in a new discussion on this technique within the introduction (line 70).

11. Lines 39-42: Im not sure anyone thinks that metabolic exchange alone could account for the degree of biodiversity observed in nature. It is simply one of many mechanisms that could promote biodiversity. An open question is whether it plays a minor or predominant role in maintaining and promoting biodiversity. Restate?

We agree with the reviewer's point and have rephrased this section of the Introduction (lines 35-39).

12. Lines 44-45: "radically different mechanism". I think this is an overstatement. Research regarding costless byproduct cross-feeding predates research regarding costly cross-feeding. The recent shift towards investigating costly cross-feeding was

largely to address social evolutionary questions, not to propose it as more or less prevalent than costless byproduct cross-feeding.

We thank the reviewer for bringing this to our attention. It was not our intention to overstate the impact of studying costless byproduct cross-feeding, and have changed the wording of the section (lines 41-51).

Reviewer #2 (Remarks to the Author):

The manuscript by Pacheco et al. uses metabolic models to chart “costless” metabolite secretions that can lead to inter-species interactions such as cooperation (beginning of which is still an open question in ecology). This question has previously been tackled, also using metabolic models, by several publications (including landmark papers from the same group). Compared to the previous studies, this work considers multiple nutrient contexts (highly relevant from ecological perspective), explicit cost on growth, and looks into different possible interaction motifs and their stability in a steady-state (simulated as chemostat) context. Overall, the manuscript provides a comprehensive perspective on possible metabolite exchanges in microbial communities. This comprehensive analysis is a major advance in understanding the broad role that metabolic interactions play in microbial communities. There are few comments (mostly semantics, missed discussion points etc.) that I recommend authors to consider for revising the manuscript.

We thank the reviewer for the positive comments, and appreciate the constructive points raised.

1. The use of “radically different” is not be justified in my opinion since the idea of costless secretion as beginning of cooperation is not new in ecology. Also see my next comment.

We agree with the reviewer that our use of “radically different” may have given the impression of disregarding previous work on costless byproduct cross-feeding. It was not our intention to do so, as we aimed instead to highlight how the magnitude of how this mode of exchange contributes to interspecies interactions and taxonomic diversity remains poorly understood. We have modified the section to better reflect this angle (lines 41-51).

2. The term costless is not accurate in the context of this study.

We thank the reviewer for the concerns raised in relation to this point, which we have addressed with multiple modifications to the manuscript:

- a. The analogy of lion->vulture is not applicable in the case of metabolite secretion.

We agree with the reviewer that this analogy may not be suitable as a direct comparison to the phenomenon we describe, and have therefore removed it.

- b. Excretion of metabolites, even if they are overflowing, do have a cost in terms of production as well as transport (both protein mediated or passive secretions take some resources; even if little, it can have fitness implications). These resource requirements are not accounted by the metabolic models (at least not by the kind used in this study).

This is an excellent point and we thank the reviewer for giving us the opportunity to clarify our definition of ‘costless.’ We wholly agree that all metabolic processes impose inherent costs to the organism undertaking them. These costs are associated with processes ranging from the generation of ATP, to the assembly of membrane transporters, to the transcriptional and translational machinery that encode for metabolite synthesis. These underlying processes are largely incorporated into the genome-scale metabolic models that we use in the study in the following ways:

- i. *The genome-scale models contain a reaction that accounts for the energy (in ATP) necessary for the synthesis of all macromolecules required for growth, such as proteins and nucleic acids. This reaction, usually abbreviated ‘GAM’ for ‘growth-associated ATP maintenance,’ has a lower flux bound that is usually determined experimentally (PMID: 20057383, PMID: 8368835, PMID: 18623053).*
- ii. *The models also contain a reaction that accounts for ATP usage for processes that are not needed for growth. This reaction, commonly called ‘NGAM’ for ‘non-growth-associated ATP maintenance,’ constitutes an ATP hydrolysis step ($1 \text{ ATP} + 1 \text{ H}_2\text{O} \rightarrow 1 \text{ ADP} + 1 \text{ P}_i + 1 \text{ H}^+$) whose constraints are also determined experimentally.*

Thus, in order for the organism to grow in silico, it must first fulfill the requirements of both these reactions.

*Nonetheless, even though the genome-scale models we use in our method account for these inherent costs, our definition of a ‘costless’ metabolite depends on the fitness burden that metabolite places on the organism. More formally, if biosynthesis and secretion of a metabolite were costly by our definition, then the organism secreting it would grow at a slower rate than if it were not. In contrast, secretion of a costless metabolite would not cause a reduction in growth rate. For example, we find that *S. cerevisiae* is able to secrete L-alanine at no detriment to its growth rate in 141 anaerobic simulations, when certain carbon source pairs are provided (e.g. citrate + glucose, glucose + malate, combinations*

of some amino acids). Under all other environmental conditions, L-alanine secretion may be possible, but not while maintaining an optimal growth rate.

We have added a new section to the introduction which more clearly defines this definition (line 58), as well as a section in the methods that more clearly define the role of GAM and NGAM in the models (line 632).

- c. Secondly, overflow metabolism is often due to some benefit to the organism, e.g. rapid substrate depletion / fast growth (e.g. ethanol secretion by yeast – see e.g. Nilsson and Nielsen, 2016) or avoiding toxicity (e.g. N-overflow by yeast – see e.g. Ponomarova et al., 2017). This is why usually the cost is carried by the secretors.

We agree with this important point and wish to clarify that this mode of secretion is encompassed in our modeling strategy. As mentioned previously, if no reduction in fitness is associated with secretion of a metabolite, then it is deemed ‘costless’ by our definition. Any inherent metabolic expenditures (ATP, transporters) are accounted for in the models and applied to all secretions, costly or otherwise.

3. The term cooperation in ecology is often used to imply evolution for that specific purpose. Thus, a better term to use here is mutualism, unless and until the authors can show evidence for evolutionary selection.

In our study, we have defined the terms ‘commensalism’ and ‘mutualism’ to signify a unidirectional and a bidirectional exchange of resources, respectively. We nonetheless understand that our use of the term ‘cooperation’ may be confusing in the context of interactions that are not evolved, so we have changed the word throughout the manuscript to ‘beneficial’ or variations thereof.

4. The secretion of AA /other metabolites can be highly regulatory context dependent – in case of yeast it requires abundant N availability and signal for NCR de-repression (e.g. poor-quality N source or perturbation of TOR pathway). These effects are not accounted for by the metabolic models used. This limitation should be discussed in the manuscript.

We agree that this is an important limitation to address. We have added an expanded enumeration of the factors that can influence metabolite secretion (including regulation), along with key references in the Discussion (line 582).

5. The methods section does not specify what is the maximum oxygen uptake rate imposed under aerobic conditions. Usually there is no secretion of, e.g. the classic overflow metabolites in the model (acetate, ethanol etc.), unless one simulates overflow metabolism by artificially limiting the maximum oxygen uptake rate (or using molecular

crowding or other resource allocation considerations). The authors should be clearer on the choices made here.

We thank the reviewer for allowing us to clarify this point. We did notice that in earlier runs of the simulations (not reported), organisms growing aerobically seemed to secrete an excessive amount of central carbon metabolism intermediates. We realized this was due to setting the maximum oxygen uptake rate too low, which was leading to overflow-like patterns of secretion. The results reported have an unconstrained oxygen uptake reaction, and we have clarified our decision in the Methods section (line 735).

6. Something seems wrong with Figure 7c (maybe the order of the plots is incorrect). I would expect growth-coupling of species 2 to species 1 in cases C1a and M1a. On the other hand, in M1b I would expect a symmetric plot since the topology is also symmetric.

In the case of C1a and M1a, the rate of byproduct secretion on the part of organism 1 is enough to sustain growth of organism 2, even for high specific growth rates for organism 2. This is because even though the maximum specific growth rate for organism 2 may be high, the effective growth rate is scaled down depending on the concentration of the byproduct on which it depends. As such, when the system reaches equilibrium, organism 2 has reached a population level such that its rate of byproduct consumption along with the dilution rate equal the rate of byproduct secretion by organism 1. These results suggest that competition for primary resources (i.e. carbon sources) is a greater determiner of the possible space of stable solutions in this continuous culture model. We have added a clarification of this phenomenon in the Results section (line 495).

We have also expanded Figure S9 to show a more fine-grained perspective on the dynamics of an individual motif.

In the case of M1b, though the topology is indeed symmetric, our selection of initial nutrient concentrations did not accurately reflect the behavior of the motif. We thank the reviewer for bringing attention to this issue and have clearly outlined our selection of parameters and initial conditions in Supplementary Table 4. The updated plot for M1b is symmetric in accordance with the topology.

7. The simulated chemostat experiments are unclear. The (pseudo) steady-state growth rates in a chemostat need equal the dilution rate, so how two species can have different growth rates in a chemostat?

We thank the reviewer for pointing out this lack of clarity. The parameter we varied is the maximum specific growth rate of the organism, or μ_{max} in our chemostat equations. We realize that the term 'growth rate' is meant to refer to the entire portion of the chemostat equation that is not related to the dilution rate which, for steady state growth, must indeed equal the dilution rate. We changed the language throughout the manuscript to clarify this point (Results line 479, Methods line 823).

8. Along the paper the authors often mention cases where predicted interactions match known interactions from the literature. These cases should be presented together into a supplementary table.

The majority of mentions of this type in the paper concern predicted secretion patterns being mirrored in the literature. We have compiled the relevant references to these, as well as to predicted interactions in a supplementary table as recommended by the reviewer (Supplementary Information 5).

Reviewer #3 (Remarks to the Author):

The present study assesses the landscape of cross-feeding interactions that can arise in a generic set of facultative anaerobic microbial organisms across many environmental conditions. The authors use an exhaustive modelling approach based on full metabolic models of the species to determine the potential of metabolites to arise and be secreted costlessly and subsequently promote one-directional and bidirectional ecological interactions. The main hypothesis states that costless metabolites drive the emergence of microbial community interactions.

Simulations suggest that the cost of metabolite 'production' varies with environment, in particular with alternative organisms and produced metabolites being present in the microenvironment. The obtained results indicate oxygen availability as strong influencer of variety (decreasing) and number (increasing) of costless metabolites generated and the number of potential mutualistic (decreasing) interactions. Furthermore they find that oxygen availability also drives patterns of co-occurring metabolite production that overall enhance growth capabilities for the present organisms. The authors link their results with ecological theory for explaining stable biodiversity and show that synergistic mutualistic interactions (i.e. positive feedbacks) are the main driver of system stability, even if the organisms compete for resources at the same time.

The study is well designed, uses cutting edge modelling and presents a very thorough interpretation of the results with the important objective to contribute novel insights to microbial community ecology. Understanding the drivers of community robustness and biodiversity in microbial communities (MCs) is of highest importance for basic research as well as for applications in public health, biotech or climate change research. The simulations also demonstrate clearly how the cost of metabolite production is a function of the environment and hence robustness of metabolic interactions based on costless production should also be. There are however some shortcomings that are of technical nature that limit the impact of the study.

We thank the reviewer for the positive assessment of our work and for the valuable insight.

1. First and foremost, the choice of organisms is based on availability of curated metabolic models as discussed in the text. This is understandable from a technical point of view, but inevitably raises the question whether studying the combinatorics of

potential metabolic overlaps is meaningful for real scenarios. More profoundly, the question arises whether real communities would show distinct patterns due to adaptive processes within the habitat. As no additional material is presented that can address this aspect (e.g. case study for particular habitat, clear binning of organisms in the used data set which likely live together or which have exclusive habitats), the main argument is ultimately not conclusive. Based on the evidence, it cannot be decided whether observed patterns either represent the natural (random) background or some fundamental rules of microbiome evolution. The expectations raised by the exciting study setup are unfortunately missed and sold below value.

We thank the reviewer for this excellent point. Though our choice of organisms was indeed limited by the availability of curated and experimentally-verified models, we sought nonetheless to incorporate organisms that come from diverse taxa and that use varied metabolic strategies. Though we believe that this choice contributes to the generalizability of our results, we do recognize that it will be essential to continue to test and refine our results as new curated genome-scale models are published.

Our choice of genome-scale models also includes organisms that are commonly used for in vitro laboratory studies of metabolism and cross-feeding. We therefore believe that our analysis is well-suited to predicting metabolic exchange and beneficial interactions in laboratory settings, which can contribute to the generation of synthetic ecologies irrespective of species' coevolution or co-localization in nature. Moreover, another factor impacting our choice of organisms is their ability to grow with and without oxygen, in order for us to directly compare the effect of oxygen on growth, costless secretions, and cross-feeding patterns.

Nonetheless, we do appreciate the importance of approximating the taxonomic and environmental contexts that organisms face in nature. To address the reviewer's comment, we have carried out a new set of simulations with organisms binned by natural environment. These simulations are separated into three sets: a set with organisms from soil environments (with oxygen, 450,684 simulations), from aquatic environments (with oxygen, 381,348 simulations), and from human gut environments (without oxygen, 381,348 simulations). These simulations feature 13, 12, and 12 organisms (Supplementary Information 1) for each respective environment.

We found that, similar to our non-habitat-specific results, exchange of costlessly-secreted metabolic products allowed for substantial increases in the ability of organisms to survive (increases in growth-supporting environments of 65.5% in aquatic habitats, 55.5% in soil habitats, and 50.7% in gut habitats). Organisms from aquatic and soil habitats had metabolite secretion profiles that were on the whole more similar to our core results with oxygen, with inorganic compounds making up the majority of secretions followed by a smaller range of organic acids and peptides. However, organisms from the gut habitat secreted a substantially higher number of unique peptides. We report these results in the text (line 325) and in a new Figure S7.

Competition phenotypes for aquatic and soil habitats again mirrored those of our core results with oxygen (majority of organisms competing for one or both carbon sources), though non-competitive, commensal phenotypes dominated the results for gut-associated organisms. This may be due to the widespread secretion of amino acids causing recipient organisms to unidirectionally depend on costless secretions and not compete for primary carbon sources, despite the lack of oxygen. These results are also discussed in the text (line 441).

While we are encouraged by the general similarities of these results to our core dataset, we lean toward interpreting these habitat-specific predictions with caution. This is especially true for comparing conclusions that have to do with oxygen availability, such as the prediction that a lack of oxygen may promote the rise of mutualistic interactions. As there is no way to directly predict how the aerobic soil and aquatic habitat-associated microbes would grow anaerobically, or how the anaerobic gut microbes would grow with oxygen, it is difficult to use these smaller simulation sets to draw generalizable conclusions on interactions and competitive phenotypes. We therefore added a note of caution on interpreting these results in the text (line 453). Nonetheless, it is possible to observe how the exchange of costless metabolites, independent of oxygen availability or habitat, can substantially increase the ability of minimal environments to support growth.

2. A second shortcoming, which is also pointed out by the authors, is the lack of including microbial pH requirements into the model. The model simulates the production of metabolites that are thereafter available in the medium and can serve to support biomass production of interacting organisms. As medium composition is known at each point in the simulation, potential pH changes should be straightforward to approximate. These would have the advantage of contributing realistic boundary conditions for individual growth dynamics and these would reduce the otherwise unconstrained interaction space.

We agree with the reviewer that this is an important consideration to address. As the composition of the medium is indeed known throughout all the simulations, it is possible to identify the presence of potentially pH-changing compounds that may detriment growth. Candidate simulations include those that feature secretion of organic acids, which at high concentrations may lead to acidification of the environment. It is nonetheless extremely difficult to make quantitative assumptions past this point, and even the capability of a compound to change the pH of the environment depends on external factors such as the buffer capacity of the medium and atmospheric composition. Moreover, though recent work has shown the dramatic effects of acidification in microbial ecosystems (PMID 29662223), a mechanistic connection between metabolic secretions and changes in pH remains unclear.

Flux balance analysis has previously been used to estimate the direction of pH change (PMID: 12952533), but direct correlation between metabolite flux and quantification of pH remains difficult to determine. As part of the revisions to our work, we performed a limited number of additional simulations using dynamic flux balance analysis (dFBA),

which yielded the concentrations of the metabolites secreted into the medium (Results line 182 Methods line 840). dFBA has previously been used to analyze culture growth in the presence of pH constraints (PMID 25519981), but this was performed under a single, well-defined environmental condition. Given the space of our simulation set, we feel it remains outside the scope of this computational study to determine the buffer capacity of each environment we tested as an approximate initial pH-based constraint. There is nonetheless a different project in our group that strives to connect dFBA-derived metabolite concentrations and changes in pH, though for similar reasons it is constrained in scope to single organisms in simple environments.

It is not our intention to downplay or ignore the effects of pH on microbial growth and metabolic exchange. However, estimation of pH at each step in our simulations unfortunately remains far outside the capabilities of the modeling techniques we have employed. To address this limitation, we have expanded the discussion of the boundaries of our modeling technique in the Discussion (line 582)

3. To exclude that variation in metabolite production is driven by oxygen availability or by the organisms the authors present a PCA analysis. Unfortunately the figure is over-plotted, too small and lacking important information with regards to the first principle components composition (axes!) that would allow to evaluate this conclusion. Indeed, a direct strategy to assess effect sizes of potential drivers (oxygen, species, interactions) would much better support this statement. For example principal component regression or MANOVA provide objective, comparable measures that enable ranking and conclusion.

We thank the reviewer for the opportunity to improve our data analysis and reporting. Through this PCA, we intended to show that not one single variable (oxygen, species, carbon source) was solely responsible for the variability we observe in each simulation's set of costlessly-secreted products. We have, however, recognized the limitations of our principal component analysis in quantitatively demonstrating the effect sizes of these potential drivers. After careful consideration of methods, we elected to use a machine learning approach based on support vector machines (SVMs) to carry out this analysis. SVMs allow for prediction of a class (in our case oxygen availability, the identity of the secreting organism, or the type of carbon source consumed) from a multidimensional dataset (in our case metabolite secretion profiles). Most importantly, this method allows us to quantitatively estimate the degree to which a potential driver influences a response. We have outlined our process of SVM training and validation in the Methods (line 828).

We found that oxygen availability and carbon source type can be considered good determiners of secreted metabolites, with cross-validation accuracies of 93.4% and 85.3% respectively. We were also initially surprised to see such a poor accuracy for organism identity (58.0%). We nonetheless believe this is because, though an organism may have the metabolic pathways for secreting a particular byproduct, such a pathway may not be active unless the required substrates are present. This machine learning

analysis and result mirrors recent experimental tests published by our group (PMID: 30072533), and is discussed in the text (line 168).

As cross-validation accuracy is the primary statistic we report from this analysis, we have directly incorporated it into the text and removed the PCA supplementary figure.

4. Finally, the authors miss to set their study in context with the scientific field, including work published by Elhanan Borenstein, Shiri Freilich, Corina Tarnita or broader, symbiosis research.

We thank the reviewer for bringing attention to this point. We have added several sentences to the Introduction and Discussion that contextualize our study in light of important advancements made by the suggested authors, among others (Introduction lines 48 and 70, Discussion line 582).

Minor comments:

5. One of the main outcomes is that mutualistic, positive interactions promote stable coexistence, compared to unidirectional commensalism – in other words increasing positive dependence enables stability. I find this is a very interesting theoretical result that should be given more focus in the presentation. It is in line with earlier results from the lab, yet should be discussed with respect to the idea that negative interactions contribute robustness (in spatial explicit systems).

We thank the reviewer for this insight and have expanded our discussion of this predicted phenomenon in the Results (line 495). We have also contextualized this observation in light of work performed by our group and other researchers (Discussion line 550).

6. From the introduction I get the impression that the authors make a categorical distinction between selfish, unwanted and other interactions. I recommend modifying the section in accordance with current evolutionary theory.

It was not our intention to make a categorical difference between these modes of exchange, as interactions mediated by costly and costless secretions essentially represent the same interaction phenomenon (cross-feeding). We have re-worded the introduction to more appropriately frame our definition of ‘costless’ and to better contextualize metabolic exchange driven by costless secretions.

7. The central definition of costless production of metabolites requires a clearer explanation and I suggest to present the underlying math in the main text to support the definition.

We have added an explicit definition of ‘costless’ in the Introduction (line 58). Additionally, we have incorporated the key mathematical variables behind our definition

into our formal exploration of the definition of ‘costless’ in the first section of the Results (line 90).

8. In several figure captions the text is still missing (fig3, fig 6, fig S6, fig S7, fig S8).

We have added extended captions to the figures referenced by the reviewer. The ordering of some supplementary figures has been changed as part of the revision: Fig. S7 is now Fig. S8, and Fig. S8 is now Fig. S9.

9. Supplement 4 is referenced in the manuscript, but not available.

We apologize for the difficulty in accessing the Supplementary Information. We wish to clarify that it is contained in the supplementary information Excel file.

10. Figure 1: Oxygen logos are too small to read when printed

The figure has been modified to improve the visibility of the logos.

11. L145 typo: a *positive* shift from *oxic to anoxic*

This typo has been corrected (line 158).

12. L163: presenting information on the individual metabolite shifts would be very valuable

We have compiled the results of our entire simulation set into a MATLAB file, which has been made publicly available on our Github repository (github.com/arpacheco/CostlessExchange). Contained in this repository are instructions for readers to interpret the results, as well as an explanation of the variables within. Using this data file, readers can identify detailed, simulation-specific information about which conditions and organisms gave rise to a secreted metabolite or an interaction. We apologize if we have misinterpreted the reviewer’s recommendation, but would greatly appreciate clarification if necessary.

13. L293: How is this measure of cooperative potential distinct and advantageous from other measures used (e.g. Freilich et al 2011 Nat Comm 589)?

In this section of our work, we quantify the ways in which combining carbon sources may yield improved growth in individual organisms. While Freilich et al. develop an excellent methodology to examine cooperative potential between organisms, this portion of our analysis focuses solely on the ways carbon sources may “cooperate” with each other to impact growth. This phenomenon may be compared to cooperative binding of proteins, in the sense that nonlinear effects are encountered with linear changes to inputs. That is, adding another carbon source may drastically and nonlinearly improve the abilities of one carbon source to sustain growth.

14.L413: Emergent properties are considered higher order phenomena that are driven by low-level interactions.

We have modified this sentence to clarify this point in accordance with the reviewer's comment (line 526).

Reviewers' Comments:

Reviewer #1:

Remarks to the Author:

This is a great revision that addresses all of my previous criticisms.

Reviewer #2:

Remarks to the Author:

Though the authors have tried to address my comments to the original submission, the changes made are rather cosmetic and not addressing the core underlying question of the cost of metabolite secretions: I am aware that the metabolic models involve the so-called maintenance cost. This, however, is not a universal number – rather far from it. It changes from species to species (or from cell type to cell type) as well as in response to changes in environment or in general physiological condition. My comment more particularly questions the costs for synthesizing the secreted metabolites, costs associated with the regulation, transporter, cell crowding etc. Some of these aspects can indeed be modelled, e.g. by constraining overall protein abundance, by considering membrane space constraints etc. The models used in these studies do not account for such constraints. The result being that the term “costless” is not on a firm basis and can hugely mislead the field. In reality, there will thus always be a cost for metabolite secretion. The metabolite secretion can be beneficial, but that does not mean that it has to be costless. The associated benefit simply needs to outweigh the costs.

In conclusion, the assertion made by the author in their response letter – that the maintenance reaction accounts for these complex situations – is incorrect and this point needs more deep addressing. At the minimal, the claim for “costless” should be removed and replaced with more accurate terminology.

Reviewer #3:

Remarks to the Author:

The revised study is a remarkable piece of microbiome modeling that provides not only fundamental insight on the context-dependent landscape of cross-feeding interactions and evolution, but hands-on information for research in gut, aquatic and soil environments. Importantly, the revised environmental stratification provides important additive value to the study.

For this manuscript version I provide few minor comments and follow up on earlier comments:

Ad comment 12 L163: I apologize if the earlier comment was stated in an unclear manner, the added supporting tables cover my comment to large extend, the missing parts are stated in the following:

Can you add a translation table of the used IDs for metabolites, reactions and genes directly in ST1 (to better understand columns L,M,N).

The matlab file is comprehensive, but since matlab has a proprietary license it is not accessible for the general public. Please additionally provide the central information in open license format.

Can you additionally provide a simple summary list with metabolites ranked by highest exchange likelihood for the individual habitats (aquatic, soil, human gut). This would enable targeting them in other studies and hence provide high additional value.

Ad comment 13: Please add a clarifying sentence about this study's focus on carbon sources in the introduction, eg. in the paragraph line 41

Supplementary information:

Figure 6, 8 the axis information is too small for reading. Please increase size or provide a readable list in the same appearance sequence at the side.

We are grateful to the editors and reviewers for the additional valuable comments on our manuscript. We feel that our manuscript has greatly improved as an outcome of the review process. All further edits requested as described below.

Reviewer #1 (Remarks to the Author):

This is a great revision that addresses all of my previous criticisms.

We thank the reviewer for all the insightful comments, and are glad that our revised manuscript satisfactorily addressed the points raised.

Reviewer #2 (Remarks to the Author):

Though the authors have tried to address my comments to the original submission, the changes made are rather cosmetic and not addressing the core underlying question of the cost of metabolite secretions: I am aware that the metabolic models involve the so-called maintenance cost. This, however, is not a universal number – rather far from it. It changes from species to species (or from cell type to cell type) as well as in response to changes in environment or in general physiological condition. My comment more particularly questions the costs for synthesizing the secreted metabolites, costs associated with the regulation, transporter, cell crowding etc. Some of these aspects can indeed be modelled, e.g. by constraining overall protein abundance, by considering membrane space constraints etc. The models used in these studies do not account for such constraints. The result being that the term “costless” is not on a firm basis and can hugely mislead the field. In reality, there will thus always be a cost for metabolite secretion. The metabolite secretion can be beneficial, but that does not mean that it has to be costless. The associated benefit simply needs to outweigh the costs. In conclusion, the assertion made by the author in their response letter – that the maintenance reaction accounts for these complex situations – is incorrect and this point needs more deep addressing. At the minimal, the claim for “costless” should be removed and replaced with more accurate terminology.

We thank the reviewer for the helpful remarks and welcome the opportunity to further clarify our terminology, as recommended by the Editor. We agree with the reviewer that there are multiple biochemical considerations that affect the cost of producing the cost of a metabolite which are not captured explicitly by FBA. We had already made references to some of these factors in the Discussion in the previous revision (Line 443), and have added a further qualification early in the Results section as requested by the Editor (Line 89). In this section (Line 73), as well as in an expanded form in the Introduction (Line 46), we have further clarified our definition of ‘costless’ and commented on the multiple limitations of our method. In addition, we now make it clearer that our definition of costless is indeed determined, as mentioned by the reviewer, by the net balance between benefits and costs, as reflected in the organism’s growth rate: if the growth rate of an organism secreting a metabolite is greater than or equal to its growth rate when it does not secrete the metabolite, then we deem this metabolite to be ‘costless’ within the tested environmental context. We wish to further

clarify that while maintenance constraints are parts of our FBA calculations, our calculations of costless secretions do not hinge specifically on maintenance flux values.

Reviewer #3 (Remarks to the Author):

The revised study is a remarkable piece of microbiome modeling that provides not only fundamental insight on the context-dependent landscape of cross-feeding interactions and evolution, but hands-on information for research in gut, aquatic and soil environments. Importantly, the revised environmental stratification provides important additive value to the study.

We are grateful to the reviewer for the positive remarks and helpful recommendations.

For this manuscript version I provide few minor comments and follow up on earlier comments:

Ad comment 12 L163: I apologize if the earlier comment was stated in an unclear manner, the added supporting tables cover my comment to large extend, the missing parts are stated in the following:

Can you add a translation table of the used IDs for metabolites, reactions and genes directly in ST1 (to better understand columns L,M,N).

The columns referenced by the reviewer denote the number of metabolites, reactions, and genes contained within the genome-scale model. We have clarified the headings to reflect these quantities accurately.

The matlab file is comprehensive, but since matlab has a proprietary license it is not accessible for the general public. Please additionally provide the central information in open license format.

We have added a CSV version of our data file to our Github repository (github.com/arpacheco/CostlessExchange).

Can you additionally provide a simple summary list with metabolites ranked by highest exchange likelihood for the individual habitats (aquatic, soil, human gut). This would enable targeting them in other studies and hence provide high additional value.

We have added an additional supplementary table (Supplementary Table 6), which lists each metabolite predicted to be exchanged in each environment, in descending order of frequency.

Ad comment 13: Please add a clarifying sentence about this study's focus on carbon sources in the introduction, eg. in the paragraph line 41

We have added a clarifier in the Introduction to better highlight the focus of this study (Line 50).

Supplementary information:

Figure 6, 8 the axis information is too small for reading. Please increase size or provide a readable list in the same appearance sequence at the side.

We thank the reviewer for pointing out this difficulty. We have doubled the size of Supplementary Figure 6 for improved visibility. As the amount of information displayed within Supplementary Figure 8 is too large to be appropriately read, we have created an additional supplementary table (Supplementary Table 7), which contains the carbon sources in the order in which they appear in the figure for improved legibility.